

# On the intrinsic time-scales of temporal variability in measurements of the surface solar radiation

Marc Bengulescu, Philippe Blanc, and Lucien Wald

MINES ParisTech, PSL Research University, Centre for Observation, Impacts, Energy
CS 10207 - 06904 Sophia Antipolis Cedex, France

*Correspondence to:* M. Bengulescu (marc.bengulescu@mines-paristech.fr)

**Abstract.** This study is concerned with the intrinsic temporal scales of the variability of the surface solar irradiance (SSI). The data consist of decennial time-series of daily means of the SSI spanning ten years, obtained from high quality measurements of the broadband solar radiation impinging on a horizontal plane at ground level, issued from different Baseline Surface Radiation Network (BSRN) ground stations around the world. First, embedded oscillations roughly sorted by ranges of increasing time-scales of the data are extracted by empirical mode decomposition. Next, Hilbert spectral analysis is applied to obtain an amplitude modulation – frequency-modulation (AM–FM) representation of the data. The time-varying nature of the characteristic time-scales of variability, along with the variations of the signal intensity, are thus revealed. A novel, adaptive null-hypothesis based on the general statistical characteristics of noise is employed, in order to discriminate between the different features of the data, those that have a deterministic origin and those being realisations of various stochastic processes. The data has a significant spectral peak corresponding to the yearly variability cycle and features quasi-stochastic high-frequency "weather noise", irrespective of the geographical location or of the local climate. Moreover, the amplitude of this latter feature is shown to be modulated by variations of the yearly cycle, indicative of non-linear multiplicative cross-scale couplings. The study has possible implications on the modelling and the forecast of the surface solar radiation, by clearly discriminating the deterministic from the quasi-stochastic character of the data, at different local time-scales.

**Keywords.** solar radiation; temporal variability; Hilbert-Huang transform; empirical mode decomposition; Baseline Surface Radiation Network (BSRN); fractional re-sampling; stochastic components.

## 1   Introduction

The power of the electromagnetic radiation from the Sun that reaches the surface of the Earth is estimated at around $10^{17}$ W. Thus, solar irradiance is the main driver behind the weather and climate systems on the planet. As such, the Global Climate Observing System (GCOS) program has identified the surface solar irradiance (SSI) as an Essential Climate Variable that helps understand climate evolution and guides adaptation and mitigation efforts (Bojinski et al., 2014). Long term time-series of the SSI are instrumental in engineering and finance by enabling, e.g. the optimal determination of geographical sites for solar power plants and guiding investment decisions, respectively (Schroedter-Homscheidt et al., 2006). Thus, better knowledge of the SSI and of its temporal variability, as recorded in long term time-series, is one of the intents of this work.





Temporally, the SSI exhibits a very wide dynamic range. Its short-term time-scales of variability, such as clouds briefly obscuring the Sun, are measured in seconds. At the opposite scale thousands or even millions of years are to be used, as related to the change of the orbital parameters of the Earth-Sun system or to stellar evolution (Beer et al., 2006). In spite of this large span of characteristic scales of temporal variability, most of the studies dealing with this physical quantity have focused

primarily on a few selected time-scales of interest. As such, reports have either dealt with global averages and long-term trends (Trenberth et al., 2009; Wahab et al., 2010; Pachauri et al., 2014; Blanc et al., 2015), have only scrutinized the short-term, high-frequency variability (Yordanov et al., 2013; Lauret et al., 2016), or have focused exclusively on a few intermediate scales (Coskun et al., 2011; Medvigy and Beaulieu, 2012). Although considerably differing in methods, taken together the previously cited studies add valuable contributions to our knowledge of the SSI. But is it possible to analyse the variability of the SSI

across multiple time-scales in a unitary way?

To do so, first a decomposition of the time-series into uncorrelated sub-constituents with distinct characteristic time-scales should be preferred. Analysis would then ensue in a like manner for each scale. The time-scales, or characteristic periods of a time-series can be identified with the inverse of the frequency at which the processes that generate the data occur. It then follows that methods portraying the changes of the spectral content of a time-series with respect to time are potentially good candidates.

This would enable both the identification of the periodicities and of the dynamic evolution of the processes generating the data. A general class of useful signal processing techniques can thus be identified in the so-called time-frequency distributions, that depict the intensity (or energy) of a signal in the time and the frequency domains simultaneously (Cohen, 1989). Such methods are commonly employed for geophysical signal processing (Tary et al., 2014).

Another factor to be taken into account are the non-linear and non-stationary characteristics of the measured solar radiation

data (Zeng et al., 2013). Handling such data issued from the non-linear interaction of physical processes, often also found under the influence of non-stationary external forcings calls for an adaptive data analysis approach (Wu et al., 2011).

The study at hand will make use of the Hilbert-Huang Transform, an adaptive, data-driven analysis technique designed specifically for investigating non-linear and non-stationary data (Huang et al., 1998). The HHT adaptively decomposes any dataset into basis functions that are derived solely from the local properties of the time-series. A time-frequency-energy rep-

resentation of the data is then constructed from these basis functions. The HHT has seen extensive use in geophysical signal analysis and spectral estimation (Huang and Wu, 2008; Huang and Shen, 2014; Tary et al., 2014). The HHT has also been previously employed on SSI datasets (Duffy, 2004; Calif et al., 2013; Bengulescu et al., 2016a, b). A similar technique has been independently proposed by Nagovitsyn (1997) for the analysis of the non-linear, non-stationary, long range solar activity. In this light, the use of the HHT for the study of the temporal variability of the SSI appears to be appropriate. The inner workings

of this data processing method will be detailed in a dedicated subsection.

Regardless of the methods used, when analysing data a question always needs to be addressed, namely: what is "signal" and what is "noise"? More precisely, there is always the need to discriminate between deterministic signals and what are assumed to be stochastic realizations of a noisy background (Rios et al., 2015). The classical way to solve this when employing the HHT on geophysical signals, such as the SSI, is to presume some model for the background power spectrum, against which the

identified features are then compared (Huang and Wu, 2008; Franzke, 2009, 2012). In contrast, the present study parts with the



traditional approach, by adopting a novel, adaptive null-hypothesis that requires no *a priori* knowledge of the background noise, introduced by Chen et al. (2013); further discussion thereof will be presented in due course. A somewhat similar objective can be found in the work of Rios and de Mello (2016), though their method of discrimination between stochastic and deterministic components is fundamentally different. Kolotkov et al. (2016) also propose a method for discriminating frequency–dependent

noise components by empirically estimating their power law spectral energy distribution and respective confidence bounds.

At this point, the general outline of our study can be summarized as follows. We analyse measurements of daily means of SSI at different geographical locations. We focus on identifying and analysing the intrinsic modes of the temporal variability of the SSI, as revealed by the HHT. We also investigate the physical and statistical significance of these modes. We show that the HHT is able to discriminate between a deterministic yearly cycle and a high-frequency (quasi-)stochastic part, that we termed

"weather noise" following Chekroun et al. (2011). Although the weather noise appears to be random in nature, we nevertheless find a non-null, statistically significant rank correlation between its amplitude envelopes and the yearly cycle. We then discuss the possible implications of our findings on the modelling and forecast of the SSI.

The study is organised as follows. Section 2 discusses the data sources and the preprocessing. In section 3 the adaptive data analysis approach will be described. Section 4 will present the results obtained, with the discussion thereof being deferred to

section 5. Conclusions and outlook are presented in section 6. Code and data availabilities are indicated in sections 7 and 8, respectively. Lastly, acknowledgements and a bibliographical list conclude the study.

## 2 Data sources and pre-processing

The data under scrutiny in this study consist of ten-year time-series of daily means of SSI obtained from high-quality measurements performed at four different locations (table 1 and figure 1). The measurement stations are part of the Baseline Surface

Radiation Network (BSRN), a worldwide radiometric network providing accurate readings of the SSI at 1 min temporal resolution and with an uncertainty requirement at $5\ \mathrm{W\ m^{-2}}$ (Ohmura et al., 1998).

The four time-series for the period 2001–2010 have been quality checked according to Roesch et al. (2011). Next, daily means of SSI were then calculated from these raw time-series only if more than 80% of the data during daylight were valid. Lastly, any isolated missing daily means were completed by linear interpolation applied to the daily clearness index, $K_T$, which

is the ratio between the daily mean of SSI and the daily mean of the total solar irradiance received on a horizontal surface at the top of atmosphere for the same geographical coordinates.

Two measuring stations are located in Europe, one in Japan, and one in North America in order to capture various climatic conditions. Boulder (herefater abbreviated in BOU) experiences a mid-latitude steppe, cool type of climate (Köppen-Geiger: BSk), while at Carpentras (abbreviated in CAR) the climate is a humid subtropical, Mediterranean one (Köppen-Geiger: Csa).

Both sites experience many sunny days during the year. As a rule of thumb, $K_T$ equal to 0.2–0.3 denotes cloudy, overcast conditions, while $K_T$ around 0.7 indicates sunny conditions. Figure 2 exhibits the histograms of $K_T$ for the four stations. One may observe the high frequencies of the greatest values of $K_T$ for BOU and CAR. The median $\widetilde{K}_T$ is equal to 0.63 for both BOU and CAR, which means that half of the days exhibit $K_T$ greater than 0.63. The climate in Payerne (PAY) is classified as





marine west coast, mild (Köppen-Geiger: Cfb), and Tateno (TAT) has a humid subtropical, east coast climate (Köppen-Geiger: Cfa). Compared to BOU and CAR, PAY and TAT exhibit more uniform histograms, with less days with cloud-free conditions, and experience more overcast and broken clouds conditions. The median $\widetilde{K}_T$ is equal to 0.47 for PAY and 0.51 for TAT. Except for TAT, which is embedded in an urban setting, the stations are located in rural environments; the local topography for BOU and TAT is flat with grassy surfaces, while for CAR and PAY the area is hilly with cultivated surfaces (BSRN, 2015).

Any further reference to seasons and seasonal phenomena shall be understood as occurring in the northern hemisphere since the stations are situated at boreal latitudes.

## 3   Adaptive data analysis

Ideally, data analysis methods should require no assumptions to be made about the nature of the scrutinised time-series, i.e. neither linearity, nor stationarity should be presumed. This is because the true character of the underlying processes that have generated the data is usually not known beforehand. Adaptivity to the analysed data would also be a sought after feature, in the sense of not imposing a set of patterns against which data would be decomposed, but rather letting the data itself drive the decomposition. This latter criterion ensures both that the extracted components carry physical meaning, and that the influence of method-inherent mathematical artefacts on the rendered picture of temporal variability is kept to a minimum (Wu et al., 2011). Since such a decomposition is only determined by the local characteristic time scales of the data, its appropriateness to non-linear and non-stationary time-series analysis is immediate (Huang et al., 1998).

### 3.1   The Hilbert-Huang Transform

The Hilbert-Huang Transform (HHT) is an adaptive data analysis technique built with the previous considerings in mind. It involves two distinct steps, the empirical mode decomposition followed by Hilbert spectral analysis. In-depth discussion of each step is carried out within the dedicated subsections that follow.

#### 3.1.1   The Empirical Mode Decomposition

The first step of the HHT is the empirical mode decomposition (EMD), an algorithmic procedure in essence, by which oscillations that present a common local time-scale are iteratively extracted from the data. These oscillatory components of the data are called Intrinsic Mode Functions (IMFs). An IMF is any function that satisfies two criteria: (1) its number of extrema and zero crossings differ at most by one; and (2) at any data point, the mean value of its upper and lower envelopes is zero. These two properties ensure that IMFs have a well behaved Hilbert transform (Huang et al., 1998). Owing to the adaptive nature of the EMD, the IMFs represent the basis functions onto which the data is projected during decomposition. This is in contrast with the Fourier or wavelet transforms where the basis functions are fixed in advance (Huang and Wu, 2008). Once all the IMFs have been extracted, all that is left of the time-series is a residue, or trend, which cannot be mathematically thought of as an oscillation at the span of the data. A sketch of the EMD algorithm is provided in algorithm 1.



---

**Algorithm 1** EMD

---

**Require:** $x(t) \in \mathbb{R}$

  Initialize IMF counter: $k \leftarrow 0$

  Initialize residual: $r(t) \leftarrow x(t)$

  **while** $r(t)$ is not monotonic **do**

    Increment IMF counter: $k \leftarrow k + 1$

    (Re)process residual: $h(t) \leftarrow r(t)$

    **while** $h(t)$ is not an IMF[a] **do**

      Find minima and maxima of $h(t)$

      Interpolate minima to find lower envelope: $L(t)$

      Interpolate maxima to find upper envelope: $U(t)$

      Find mean of envelopes: $m(t) \leftarrow (L(t) + U(t))/2$

      Remove mean of envelopes: $h(t) \leftarrow h(t) - m(t)$

    **end while**

    Store IMF: $c_k(t) \leftarrow h(t)$

    Update residual: $r(t) \leftarrow r(t) - c_k(t)$

  **end while**

  **return** $c_{1...N}(t), r(t)$

---

[a]See text for the definition of IMF. This "sifting" loop should be run approximately 10 times (Wu and Huang, 2009, 2010).

One of the drawbacks of the original EMD is that it may introduce a phenomenon known as "mode mixing". This is the manifestation of oscillations with dissimilar time-scales in the same IMF, or the presence of oscillations with similar time-scales in different IMFs. A workaround was proposed by Wu and Huang (2009) with ensemble empirical mode decomposition

5  (EEMD). The idea was to run the decomposition over an ensemble of copies of the original signal to which white Gaussian noise has been added, with the final result obtained by averaging. Although the EEMD improved the mode mixing problem, the different sums of signal and noise produced different numbers of modes, making the final averaging somewhat difficult. Added to this, the reconstructed signal still contained some residual noise, and thus was not identical to the original. To overcome this situation, Torres et al. (2011) have proposed another iteration of the EMD, the complete EEMD with adaptive

10  noise (CEEMDAN). This method also decomposes the white noise into modes, along with the signal, such that at each stage of the decomposition a particular noise is added and a unique residue is computed to obtain each mode. However, the modes of CEEMDAN still contain some residual noise and sometimes spurious modes appear in the early stages of the decomposition. The next iteration of the method, the improved complete ensemble EMD (ICEEMD or ICEEMDAN), overcomes these issues by fixing the signal to noise ratio for all stages of the decomposition process (Colominas et al., 2014). The ICEEMD method

15  is that used in this study. In addition, a fast EMD routine provided by Wang et al. (2014) has been used to decrease the computation time.





To illustrate the workings of the EMD, the eight IMFs of the BOU time-series are presented in figure 3 in the order they were obtained, from top to bottom. As EMD operates in time-domain, the IMFs have the same temporal support as the original data and, by construction, an average of zero and symmetrical upper and lower amplitude envelopes. It can be observed in figure 3 that as the decomposition progresses, the time-scale of the IMFs increases, i.e. the intrinsic oscillations are getting further

spaced apart with increasing IMF number. Another view of this is brought by figure 4, where the power spectral density (PSD) and a Fourier estimate of the mean period of each IMF are plotted. To aid the reader, the colours used to portray the individual IMF spectra are the same as for the time-domain representation from figure 3. The spectral shapes of the IMF1...IMF5 are similar in form, i.e. bell curves, and their median periods roughly follow a dyadic scale, i.e. doubling with increasing IMF number as : 3.1 days → 7.3 days → 13.9 days → 30.5 days → 54.0 days. This doubling of the time-scale for these first five

IMFs is the hallmark output of an efficient dyadic filter. Subsequently, it will be shown that this dyadic repartition is pertinent to identifying deterministic signals from random realizations of quasi-stochastic background processes. This finding is even more interesting, since the median periods have been estimated with a Fourier-based method, which measures period *globally* over the whole time range of the IMFs. By opposition, a measure of *local* period in the Hilbert sense is a much better estimate, since it has an accuracy as low as a quarter wavelength of temporal resolution with respect to the average time-scale of the IMF

(Huang et al., 2009).

### 3.1.2 Hilbert Spectral Analysis

Once the empirical mode decomposition is completed, the second and last step of the HHT consists in the Hilbert spectral analysis of the previously obtained IMFs. Each IMF and its Hilbert transform are used to construct a complex analytic signal, described by an amplitude modulation - frequency modulation (AM–FM) model. This decomposition into two time-varying

parts corresponding respectively to instantaneous amplitude and instantaneous frequency is very useful for the purpose of this study. It enables the identification, in a time-varying sense, of how much power (i.e. the square of amplitude) occurs at which time-scale (i.e. the inverse of frequency).

The Hilbert transform of each real-valued IMF $c_k(t)$ can be written as:

$$\sigma_k(t) = \mathcal{H}(c_k(t)) = \frac{1}{\pi} P \int_{-\infty}^{\infty} \frac{c_k(\tau)}{t - \tau} \, \mathrm{d}\tau \tag{1}$$

where subscript $k$ designates the $k$-th IMF, and $P$ indicates the Cauchy principal value. From each IMF and its Hilbert-transformed version, a unique complex-valued analytic signal can be obtained (Gabor, 1946):

$$z_k(t) = c_k(t) + i \cdot \sigma_k(t) = a_k(t) \cdot e^{i \cdot \theta_k(t)} \tag{2}$$

in which

$$a_k(t) = \sqrt{c_k^2(t) + \sigma_k^2(t)} \tag{3}$$





is the instantaneous amplitude and

$$\theta_k(t) = \tan^{-1}\left(\frac{\sigma_k(t)}{c_k(t)}\right) \qquad (4)$$

is the instantaneous phase. The instantaneous frequency is the first time derivative of the instantaneous phase:

$$\omega_k(t) = \frac{1}{2\pi}\frac{\mathrm{d}\theta_k(t)}{\mathrm{d}t} \qquad (5)$$

Figure 5 provides a visual guide to this concept, by illustrating the AM–FM decomposition of IMF5 for the BOU time-series. The top panel (IMF5) of the figure reproduces the mode function, which is also the real part of the analytic signal from equation (2). The amplitude of the latter (AM), given in equation (3), which is the envelope of the original signal, is then extracted and plotted in the middle panel. Unlike in the Fourier decomposition, the amplitude is not a constant, but rather a time-dependent function. Next, by removing the AM component from the signal through simple division, the frequency modulation component

is obtained, i.e. the complex exponential in equation (2); the real part of this component (FM) is plotted in the bottom panel. The FM is similar to a trigonometric function, but with a phase argument that unlike the Fourier transform is not a constant but a time-dependent function, as seen from equation (4). The local frequency (and its inverse, the local time-scale of the signal) is then just the first temporal derivative of this phase, as defined in equation (5). Owing to their time-varying character, the amplitude and frequency are usually encountered in the literature under the terms instantaneous amplitude, and instantaneous

frequency, respectively.

The original time-series $x(t)$ can then be expressed as a sum of AM–FM signals riding onto the EMD trend, $r(t)$, as follows:

$$x(t) = \mathrm{Re}\left[\sum_{k=1}^{N} a_k(t)\cdot e^{i\int \omega_k(\tau)\,\mathrm{d}\tau}\right] + r(t) \qquad (6)$$

The square of the instantaneous amplitude and the instantaneous frequency of the IMFs can then be used to represent the

data as an energy density distribution overlaid on the time-frequency space, as in equation (7). This representation, called the Hilbert energy spectrum is defined by Huang et al. (2011) as "the energy density distribution in a time-frequency space divided into equal-sized bins of $\Delta t \times \Delta \omega$ with the value in each bin summed and designated as $a^2(t)$ at the proper time, $t$, and proper instantaneous frequency, $\omega$."

$$S(\omega,t) = \sum_{k=1}^{N} a_k^2(t)\cdot e^{i\int \omega_k(\tau)\,\mathrm{d}\tau} \qquad (7)$$

The time-integrated version of equation (7), the Hilbert marginal spectrum $S_M(\omega)$, is similar, but not identical to, the traditional Fourier spectrum:

$$S_M(\omega) = \int_0^T S(\omega,t)\,\mathrm{d}t \qquad (8)$$



An example of Hilbert spectral representation is given in figure 6 (left panel) where the BOU time-series is shown as an energy density distribution over-imposed on a time-frequency space as in equation (7). Each pixel in the Hilbert spectrum is identified by three attributes – color, abscissa, and ordinate – through which it denotes the local power (color, log-scale) of the corresponding time-series, at a certain time (abscissa), and at a certain time-scale (ordinate, log-scale). For the sake of the

readability, the spectrum is binned in time, scale, and colour space and has been smoothed. Hence, some aliasing may occur. Some features may be represented as continuous lines while others are rendered as point-like, especially where rapid frequency modulation takes place, such as in the high-frequency bands.

Interpretation of Hilbert spectral features at data boundaries must be done with care due to possible oscillations of the spline interpolants used in the EMD (see algorithm 1). This effect is similar to the "cone of influence" in the popular wavelet transform

(Torrence and Compo, 1998). With the EMD edge effects are usually contained within a half-period of a component at data boundaries (Wu et al., 2011). In figure 6 this region has been whitened out.

The plot in the right panel of figure 6 is the Hilbert marginal spectrum, or the time-integrated variant of of the image at its left, indicating the amount of power at each time-scale. This time agnostic representation is comparable, but not identical to, the Fourier spectrum of the same time-series. It should be once again emphasized that the Hilbert marginal spectrum is obtained

from *local* features of the data, its components having instantaneous amplitude and instantaneous frequency, as opposed to the *global* outlook of the Fourier spectrum whose constituents have constant amplitude and constant frequency throughout the whole domain.

## 3.2 Adaptive background noise null hypothesis

Which confidence can be attributed to the information extracted by the EMD? More specifically, knowing that measurements

may be contaminated by noise, how can one ascertain that a certain IMF is the result of a real physical process as opposed to it possibly being a stochastic manifestation of background noise?

In the past, several investigations have been carried out in order to identify the effects of the EMD when applied to time-series issued from various noise models, such as white, red, or fractional Gaussian noise, (Huang et al., 2003; Flandrin et al., 2004; Flandrin and Gonçalves, 2004; Wu and Huang, 2004; Flandrin et al., 2005; Rilling et al., 2005; Schlotthauer et al., 2009;

Colominas et al., 2012). As a result, it has been consistently shown that, irrespective of the assumed noise model, the EMD acts as an efficient "wavelet-like" dyadic filter, decomposing the noise input into IMFs having the same spectral shape, but that are shifted in the frequency domain.

Nevertheless, the rejection of a null hypothesis based on an *a priori* assumed class of noise does not preclude the probability that the now statistically significant deemed signals originate from a stochastic process of a different kind. Furthermore, as the

EMD is an adaptive, data driven decomposition, it would be desirable to also employ a null hypothesis that shares the same characteristics, making no beforehand assumptions about the character of the background noise.

Following Flandrin (2015) and Chen et al. (2013) this study will make use of the robust statistical properties of the EMD with respect to a wide class of noise models, in order to adaptively contrast potential signals against the presumably stochastic background, as detailed hereafter. Owing to its dyadic filter character, the EMD decomposes noise inputs into IMFs having

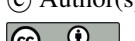



similar spectral shape, but that are translated to roughly the next lower octave in the spectral domain. When the sampling step is increased, i.e. the sampling frequency is reduced by fractionally re-sampling the input, these components cannot preserve their original locations in the spectral domain and will instead be shifted towards lower frequencies. Hence, significance testing of IMFs is done by verifying if the IMF remains unchanged in the time-frequency representation of the signal, during fractional
re-sampling of the latter.

A Hilbert marginal spectrum $S_{M_k}(\omega)$ is first constructed for each IMF from its instantaneous amplitude $a_k(t)$ and instantaneous frequency $\omega_k(t)$. Next the spectrum-weighted mean frequency (SWMF) $\overline{\omega}_k$ of each IMF is computed (Chen et al., 2013):

$$\overline{\omega}_k = \frac{\displaystyle\int S_{M_k}(\omega)\,\omega\,\mathrm{d}\omega}{\displaystyle\int S_{M_k}(\omega)\,\mathrm{d}\omega} \tag{9}$$

Then, the time-series is fractionally re-sampled by making the original sampling rate $\Delta t$ progressively larger, i.e. the time-spacing of the data points becomes:

$$\Delta t_l = \Delta t \cdot l, \quad l \in \{1.1, 1.2, \ldots, 1.9\} \tag{10}$$

For each sampling rate $l$, and for each IMF $k$, the SWMFs are then recomputed, obtaining a set $\overline{\omega}_{k,l}$. To enhance the visibility of the evolution of frequency as a function of the re-sampling rate, normalization is performed as in:

$$\widehat{\omega}_{k,l} = \frac{\overline{\omega}_{k,l}}{\overline{\omega}_{k,1}} \tag{11}$$

with $\overline{\omega}_{k,1}$ being the SWMFs of the modes of the data having the original sampling rate. Therefore, the normalized SWMFs for the IMFs of the original data will be unity, i.e. $\widehat{\omega}_{k,1} = 1, \forall k$.

Since the EMD is an efficient dyadic filter, frequency deviation from the unity line will occur for noise-like IMFs. It follows that when $\widehat{\omega}_{k,l} \simeq 1, \forall l$, the null hypothesis that mode $k$ is the realization of stochastic processes can be rejected.

**4   Results**

The IMFs obtained from the BOU time-series from figure 3 have already served as an illustrative example on the operation of the EMD. The IMFs for the other datasets (not shown) are very similar and will be discussed in due time. It must be noted that, like BOU, the CAR time-series is decomposed into 8 IMFs, while the PAY data has 9 IMFs and 10 IMFs are obtained for TAT. Besides the IMFs, for each time-series the decomposition also yields a residual, or trend (also not shown). With respect to the
decennial time span of the analysis (10 years), the trend can be thought of as a low-pass approximation of the data (Moghtaderi et al., 2013). Nevertheless, these EMD trends along with their statistical significance and physical meaning do not fall within the scope of this study; for such discussion, see e.g. (Franzke, 2012).



From the Fourier spectra of the IMFs in figure 4 it can be seen that, owing to its median period of 364.8 days, IMF6 can be unambiguously associated with the yearly cycle, as dictated by the orbital parameters of the Earth-Sun system. IMF6 also accounts for the most prominent visual feature in the original data (figure 1, top panel: BOU), with its maxima and minima denoting summer and winter, respectively. Further evidence is brought by the spectral shape of IMF6, distinguished by a sharp

peak that has the largest power in figure 4. Also noteworthy is that IMF6 seems to modulate the previous five IMFs, as these latter seem to exhibit amplitude excursions that are approximately in phase with the amplitude of IMF6, a phenomenon that is most visually distinguishable in figure 3 for the year 2005.

Finally, the last two components, IMF7 and IMF8, having median periods of 783.3 days and 1457.4 days (figure 4), respectively, are seen to exhibit only slight amplitude deviation from zero in their temporal representation. Moreover, these

fluctuations in amplitude occur at the end of the signal for IMF7 and at the front edge for IMF8. Interpretation of these components should, thus, be done with care, since edge effects for the EMD are known to be usually contained within a half-period of a component at data boundaries (Wu et al., 2011), i.e. approximately one year for IMF7 and two years for IMF8.

With the FM components obtained, it becomes possible to illustrate the frequency contents of each time-series in terms of its individual IMFs, as shown in figure 7, where by means of box plots the distribution of the local time-scale of each mode is

conveyed. This box plot representation is somehow incomplete, as it only accounts for the period distribution of the modes and does not take into account either the amplitude or the temporal localization of the events. The box plots of the instantaneous amplitude of each IMF are given in figure 8.

For all time-series, IMF1...IMF5 have very similar median periods (figure 7), that approximate the dyadic sequence: 3.5 days → 7 days → 14 days → 28 days → 56 days. Moreover, besides the notable similarity among the medians of these modes,

for all the datasets both the interquartile ranges and the total ranges of these first five modes exhibit approximately the same variability. Added to this, IMF6 for BOU, CAR, and PAY, and IMF7 for TAT, whose median periods are respectively 368.2, 364.3, 356.6, and 366.6 days, can clearly be associated to the yearly cycle given by the revolution of the Earth around the Sun. This yearly component is very similar for BOU, CAR and to a lesser extent TAT, with an interquartile range that is concentrated around almost the same median value, the only minor difference being the slightly extended range for TAT of 200

days as opposed to 300 days for the other two. The PAY yearly mode differs from those of the other stations, its interquartile range and foremost its range being much larger, the latter even overlapping the interquartile ranges of IMF5 and IMF4. This is a result of the mode mixing phenomenon described in section 3.1 that may arise with the EMD, i.e. the coexistence or mixing of different time-scales in the same IMF, mainly related to the intermittence of signal and to noise (Huang et al., 2003). Nevertheless, the spectral part of IMF6 which overlaps IMF5 and IMF4 has very low power (Bengulescu et al., 2016b), thus

this phenomenon does not influence the validity of the analysis. With this in mind, one notes that for BOU and CAR no spectral components are present in the 100 days to 300 days band. Furthermore, TAT is the only dataset that has a transitional mode of 143.2 days median period in between the first five IMFs common to all stations and the yearly cycle.

At this point, the Hilbert frequency distribution of the IMFs for BOU may be compared to the Fourier one from the PSD in figure 4. As previously mentioned, the Hilbert estimates are based on *local* features of the data, and thus are more accurate

than the Fourier ones when applied to non-stationary signals. This can be seen especially when comparing the range of the



first five high-frequency IMFs, which is upper bounded to about 100 days in figure 7, whereas in the PSD from figure 4 the spectra of the same components are seen to span the whole time-scale range. This also holds for IMF6, which has very narrow Hilbert period range, whose Fourier analogue is the sharp peak in the PSD of the same mode. Similar statements can be made for IMF7 and IMF8. To sum up, it is found that while the Hilbert period distributions of the modes have compact supports,

the Fourier representations of the same components span the whole frequency range. Nevertheless, most of the power in the Fourier PSD is assigned to a frequency band that closely corresponds to the Hilbert range. Owing to the global nature of the Fourier transform, however, additional spectral coefficients are needed to provide a complete mathematical description of the data.

Resuming the discussion of the IMF time-scales from figure 7, it can be observed that the low-frequency, i.e. greater than

one year, variability of the data, trend notwithstanding, is assigned into slightly overlapping (within the same time-series) IMFs that span the spectrum starting from the one year mark. For BOU and CAR time-series, there are only two modes extending beyond one year. First, IMF7 can be seen to span approximately the same range for both these stations, from about one year to slightly more than three years. For BOU however, the interquartile range and especially the median period is shifted towards higher periods, i.e. 724.7 days vs. 469.5 days for CAR. The last modes IMF8 of these stations are very different, with a very

narrow range around the median of 1531.5 days for BOU, and a range of 900 days to over 2000 days and median of 1305.1 days for CAR. For the PAY data, the low-frequency components have narrower spectral support, with two IMFs (IMF7 and 8) that cover the band from 1 to 2.5 years and median periods of 413.6 and 707.5 days, and the IMF9 around 4.5 years ($\backsim$ 1668 days) with a very narrow range. It must also be noted that IMF7 for BOU, CAR and PAY have the same lower end support, and that the couple (IMF7, IMF8) of PAY taken together somehow emulates IMF7 for BOU and CAR. Lastly, TAT is the

only dataset whose the low-frequency variability is expressed by three components, IMF8...IMF10, with mean periods of 609 days, 1440.3 days and 2402.6 days. While the first quartile of IMF9 coincides with the upper range of IMF8, the upper range of IMF9 is slightly below the lower range of IMF10, hence the last two modes do not overlap at all. By its range, IMF8 of TAT approximates IMF7 for BOU and CAR, but there is no proximity in terms of median or interquartile range. Similarly, IMF9 of TAT resembles IMF8 of CAR in terms of range, but their medians are not in close agreement and their interquartile ranges

even less so.

With the scrutiny of the these low frequency components, the discussion of the time-scale distribution of the IMFs from figure 7 can now be concluded. However, as previously mentioned, this particular illustration, although instructive, is incomplete. First, the box plot representation does not take the instantaneous variations of frequency into account, but renders global aggregates instead – much like the traditional Fourier methods, with the interquartile range spread in addition. This is done on

purpose, with the intent of making it easier for the readership not accustomed to the HHT to create analogies with the more familiar methods (e.g. Fourier analysis, wavelets, etc.). Second and last, this particular representation is totally devoid of any information pertaining to the local amplitude, or power, or variance, of the data. With these consideration in mind, the Hilbert spectrum, a true time-frequency representation for non-linear and non-stationary data, will be discussed next. Since the goal of this exercise is to lay the groundwork for the forthcoming discussion, only the spectrum for the BOU data will be provided as

an example.





The BOU Hilbert spectrum from figure 6 exhibits a high-frequency feature between 2 days and ⌣ 100 days, which corresponds to first five IMFs of the time-series. The instantaneous time-scales of these modes overlap (figure 7), hence the appearance on the Hilbert spectrum of a continuum instead of distinct bands. This spectral feature has relatively low power, that decreases with increasing period, as can be inferred from the sloped dent in the marginal Hilbert spectrum corresponding

to this region. In the 2 days to 32 days band, amplitude modulation by the yearly cycle can be inferred from the periodic change in color, with yellow-green tones, occurring mostly during the high irradiance regime of summer, that turn blue during the winterly minima. Next, in the band between 100 and 300 days, a gap in the spectrum is apparent, as can also be inferred from the lack of support in this region for any of the BOU IMFs in figure 7. The yellow trace, corresponding to IMF6, exhibits frequency modulation around the one year period, seen as oscillations in the range of 300 to 450 days, which is also

the support of this mode in the box plot of IMF time-scales. The colour of this IMF indicates that it has the highest power of all the components, as can also be inferred from the large peak on the marginal spectrum. The corresponding time-scale fluctuations are centred in 365 days, and frequency modulation is greatest during 2003 through 2005. From 2006 onwards, however, frequency modulation is less pronounced – perhaps capturing the low solar activity around the 2008 minimum in the eleven year cycle solar cycle (Hathaway, 2015). The final two low-frequency, blue-green traces on the spectrum correspond to

IMF7 and IMF8. For IMF7, mode mixing is apparent through the occasional sharing of the yearly time-scale band with IMF6, between mid-2003 and 2005. IMF7 has such low power that it fails to leave an imprint on the marginal spectrum and it seems to suddenly spring into existence during summer 2003, which is in perfect agreement with its temporal representation from figure 3 (panel IMF7), whence it can be seen to have negligible amplitude during the first two and a half years. Also in agreement with its temporal depiction from figure 3 (panel IMF8), IMF8 starts out in light-green hues and slowly vanishes during mid-2007.

Although this last BOU mode manages to register on the marginal spectrum through two very slight indentations around 1500 days (which is about the median period of this mode from figures 4 and 7), most of its power lies within edge effect territory, hence interpretation of these slight bumps is ambiguous at best.

Thus far, all time-series have been shown to share a high-frequency constituent between 2 days and 100 days composed of five IMFs with mean periods following a dyadic sequence, and an IMF around 365 days that captures the yearly variability. For

BOU, CAR and PAY, a low power region can be found in the 100 days to 300 days band. Beyond the one year time-scale, the low-frequency variability in the 1.5 years to 6 years band is captured by another two (BOU and CAR) or three (PAY and TAT) components. The TAT data is the only time-series that has an IMF in the low power band between the high-frequency feature and the yearly cycle (median period 143.2 days).

## 5 Discussion

The previously identified features of the SSI time-series will be now be discussed in terms of their intrinsic temporal scales of variability, and physical statistical meaning.



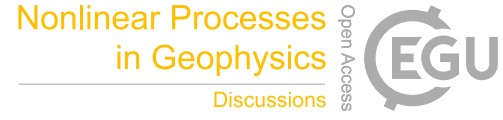

## 5.1 The intrinsic time-scales of variability of the SSI

Firstly, the median periods of the IMFs composing the high frequency band are revisited. It has been shown in figure 7 that they follow a dyadic repartition, that approximate the series dyadic sequence: 3.5, 7, 14, 28, 56 days. Such a doubling in frequency in IMFs has been previously reported in astrophysical and geophysical signals. When investigating three independent datasets

of satellite observations of the (extraterrestrial) total solar irradiance (TSI), Lee et al. (2015) consistently find a similar dyadic scale progression of modes at 13.5, 27, and 54 days, statistically significant within the 95% level, that correspond to the 27-day solar rotation period and its (sub-)harmonics. Kolotkov et al. (2015) find intrinsic periodicities having an average of $25^{+7}_{-2}$ and $44^{+10}_{-5}$ days five different solar proxy signals. The mean periods and the associated error bars in sub-/superscript, estimated at the half-level width of the corresponding probability histogram, were obtained by analysing the sunspot area for the whole Sun,

and for the northern and southern solar hemispheres taken separately, the 10.7 cm radio flux intensity, and the helioseismic frequency shift. Compelling as it may seem, nevertheless, the imprint of a solar rotation signature on ground measurements of the SSI is highly unlikely, as it would imply the existence of hitherto unknown physical mechanisms in Earth's atmosphere (Thuiller, 2015). The amplitudes of the IMFs of the TSI time-series and those of the IMFs in the SSI data differ at times by two orders of magnitude, e.g. compare figure 3 with figure 1 in (Lee et al., 2015). If the solar rotation signature were to be seen

in the IMFs of the SSI this would require the existence of amplifying processes. Stott et al. (2003) and Lockwood and Fröhlich (2007) have studied the possibility of such a mechanism, and have concluded that, irrespective of the mechanisms invoked and of the amplification of the solar variability, for the past decades solar forcing is only a minor contributor and thus not able to account for most of the global warming observed in the second half of the twentieth century, which could be better explained by an increase in greenhouse gases. Further proof will be provided subsequently, this time from a signal theoretical point of

view, in support of the view that it is unlikely that the solar rotation signature is captured in measurements of the SSI.

Secondly, in the 100 days to 300 days band, two of the stations, BOU and CAR, do not exhibit any variability. For PAY, the support of yearly IMF6 protrudes in this region, although its first quartile rests well below the 200 days mark. As mentioned before, the power of the portion of this IMF that extends into the high-frequency range is very small (not shown). Hence, while not totally devoid of spectral features, this band contains negligible power. A distinct mode is present at TAT in this band, whose

median period of 143.2 days somehow seems to continue the dyadic sequence of the previous five modes. These findings are important for the modelling and forecasting of the SSI, as follows. On the one hand, models for BOU and CAR should not containt any power in this band, or at least filter it out. For TAT, on the other hand, any model attempting to reconstruct the SSI should ensure that the 100 days to 300 days region is not a spectral void. In section 5.4, evidence will be presented that the spectral band spanning from two days to 300 days seems to be composed mostly of coloured noises, i.e. random realisations of

stochastic background processes, which can be modelled following, e.g. (Flandrin and Gonçalves, 2004; Rilling et al., 2005; Welter and Esquef, 2013; Kolotkov et al., 2016).

Thirdly, the median periods detected around the one year mark in all the datasets can be explained by the movement of revolution of the Earth around the Sun and the associated orbital parameters. The interpretation of these components is unambiguous, with one notable exception for the PAY time-series, whose IMF6 exhibits mode mixing, i.e. it has a total range that





overlaps some of the modes in the high-frequency band. Nevertheless, it will be subsequently be shown that it is indeed these components that account for variability at the one year time-scale.

Lastly, the components indicative of low-frequency variability on time-scales greater than one year are discussed. The intrinsic time-scales found in these IMFs seem to match once more those pertaining to the so-called solar quasi-biennial

oscillations, i.e. variations in the activity of the Sun exhibiting periodicities between 0.6 and 4 years (Bazilevskaya et al., 2015). Again, Kolotkov et al. (2015) identify in the modes of the five solar proxies average periods of $395^{+46}_{-46}$, $626^{+69}_{-113}$, and $1423^{+196}_{-146}$ days respectively. Vecchio et al. (2012) also report solar quasi-biennial oscillations (QBOs) with time-scales from 1 year to 4.5 as being fundamental components of the variability of solar magnetic synoptic maps. Harrison (2008) identifies a 1.68 year peak in the spectral domain by inferring cloud cover from measurements of the SSI, indicating that galactic cosmic

rays, rather than solar irradiance, may induce a cloud effect. Kirkby et al. (2011, 2016) provide some tentative evidence for such an effect. Nevertheless, the same final precautions must be reiterated, similarly to the previous discussion concerning the high-frequency constituents.

## 5.2    The local climate imprint in the IMFs

It has been shown in section 2 that the four measuring stations experience different climates and exhibit differences in terms

of $K_T$. Figure 7 shows that the high frequency band composed of the first five IMFs is very much alike for all stations. This section investigates the possible relationship between local climate and dissimilarities in terms of the repartition of the IMFs 6 and more.

It can be noted that the IMF6 for both BOU and CAR has a well-defined period (figure 7), with a median of respectively 368.2 and 364.3 days and very narrow interquartile range. In addition, for both stations, the IMF6 is the mode having the

greatest amplitude and by far, compared to the other modes (figure 8). The IMFs 7 and 8 for CAR have less marked periods, i.e. the interquartile ranges are greater than for IMF6, and the amplitude of each IMF is very small. These observations may be related to the high frequency of cloud-free days seen in figure 2 because in absence of clouds, the variability of the daily mean of SSI is predominantly driven by the variability of the solar irradiance received at the top-of-atmosphere during the year.

PAY and TAT need four IMFs to account for the low frequency variability, i.e. one IMF more than BOU and CAR. IMF6

in PAY has a median period of 356.6 days, close to one year (figure 7) with a large interquartile range. The median amplitude of the IMF6 is approximately half of that of BOU or CAR (figure 8) and the amplitude exhibits large variations. The median amplitude of the IMF7 is similar to that of IMF6 while the period of the IMF7 is well marked with a narrow interquartile range. This may be related to the abundance of the presence of broken clouds that render the SSI signal highly intermittent. This intermittence of the signal could, in turn, explain the mode mixing observed in IMF6 (Huang et al., 2003).

Similar to PAY, TAT also has a low median clearness index $\widetilde{K}_T^{\text{TAT}} = 0.51$, which helps explain the presence of a sixth IMF (median period: 143.2 days) between the high-frequency components and the yearly IMF7 (median period: 366.6 days). In other words, there is much more weather noise in the sub-year band at TAT than at PAY, or a lower signal-to-noise ratio of the yearly cycle. Hence, this high level of noise drives the EMD to assign a dedicated intrinsic mode for this region, as opposed to PAY, where the signal in this spectral band is assigned to the yearly IMF through mode mixing.



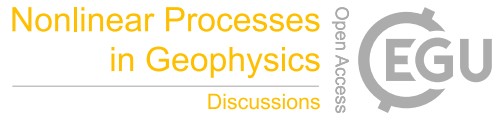

### 5.3 Discriminating signals from noise in the IMFs

At this point, having identified the spectral characteristics of the SSI time-series by means of the HHT, a question arises with regard to their physical and statistical significance, namely how can one ascertain which features represent the expression of real, deterministic physical phenomena and which ones can be attributed to random realizations of background processes. Such

a method of discriminating between the "data" and the "noise" IMFs, proposed by Chen et al. (2013), has been described in subsection 3.2. The procedure was applied to the first eight IMFs of all the time-series and the results are presented in figure 9 (from top to bottom BOU, CAR, PAY, and TAT). First, each time-series was re-sampled with a fractional sampling rate up to a factor of two, i.e. the original uniform time-spacing of the data, $\Delta t$, was progressively made larger and larger, as described in equation (10): $\Delta t_l = \Delta t \cdot l$, where $l \in \{1.1, 1.2, \ldots, 1.9\}$ is the re-sampling rate and is running along the horizontal axis.

Next, the HHT was used to decompose the resulting time-series into IMFs and to compute their spectrum-weighted mean frequencies, following equation (9). In order to emphasize the effects of the fractional re-sampling on the spectral contents of the IMFs, these latter frequencies were then normalized by the SWMF of the original, non re-sampled data as per equation (11). For each dataset, this ratio is indicated on the y-axis, as $\widehat{\omega}_{k,l}$, with $k \in \{1 \ldots 8\}$ indicating the IMF number. It then becomes possible to follow the evolution of the normalized SWMF of each individual IMF as a function of the fractional re-sampling

rate (figure 9). As the EMD is an efficient "wavelet-like" dyadic filter, it follows that the IMFs of time-series of pure noise undergo a translation towards lower frequencies under fractional re-sampling. Therefore, for those IMFs whose SWMFs are not down-shifted during re-sampling, the null hypothesis that they are pure noise can be rejected, i.e. they represent meaningful signals. Stated otherwise, an IMF $k$ is deemed not to be stochastic in nature if it normalized SMWFs $\widehat{\omega}_{k,l}$ stay close to the unity line for all $l$. From figure 9 it can be observed that for all the stations, the only component that maintains quasi-constant

frequency under fractional re-sampling is the mode representative of the yearly variability, i.e. IMF6 for BOU, CAR and PAY, and IMF7 for TAT. All the other IMFs experience the previously described frequency down-shifting, hence for them the null hypothesis that they are composed of pure noise cannot be rejected.

    At this point, several precautionary notes are compulsory. First, the rule of inference used here is *modus tollens*, i.e. the results from figure 9 do not imply that the modes who experience down-shift in their SWMFs are made up of pure noise. It will

be subsequently shown that, for the first five IMFs at least, this is indeed the case; although "noise-like", or (quasi-)stochastic in nature, they are not completely devoid of information. Second, the result is mostly qualitative, since it is difficult to define a confidence interval owing to the adaptive nature of the null hypothesis that can account for different types of noise. Third and last, the approach is best applied only to the high frequency modes, with respect to the data length and sampling, since by re-sampling spurious low-frequency oscillation may inadvertently be introduced (Chen et al., 2013). This is further supported by

the fact that as IMF number progresses, the region where the influence of edge effects becomes important is getting larger and larger, hence only adding uncertainty to the interpretation of the results. This is also the reason why this type of analysis was only carried out on the first eight IMFs of each dataset. As a corollary, unambiguous interpretations of QBO-like components seems to be out of reach.



### 5.4 Amplitude modulation through non-linear cross-scale coupling

This section investigates whether the first five IMFs can be modelled as purely uncorrelated, random noise, or if they also contain any other form of information. To test this, the rank correlation between the yearly and sub-yearly IMFs and their envelopes, e.g the AM part in the middle panel of figure 5, has been computed for each SSI time-series. Kendall's rank
correlation coefficient, $\tau$, a statistical measure of ordinal association describing how similar the orderings of the data are when ranked (Kendall, 1938), is employed here to establish whether each pair of the two variables, AM$x$ and IMF$y$ with $x, y \in \{1 \ldots 7\}$, may be regarded or not as independent. $\tau = 1$ indicates perfect agreement between rankings, while $\tau = -1$ denotes perfect disagreement, i.e. one ranking is the reverse of the other; for $\tau = 0$ the two variables are statistically independent.

    The resulting rank correlation coefficients and the associated $p$-values, are presented in figure 10. For each panel, the
columns denote the EMD modes (IMF$x$), and the rows their amplitude envelopes (AM$y$). The background colour of each cell (AM$x$, IMF$y$) indicates the rank correlation $\tau$ between IMF$y$ and the AM part of IMF$x$ within the same dataset. The legend of the colour encoding is found on the color bar at the bottom of the figure. The associated $p$-values are presented numerically in each cell, for the sake of completeness and transparency (Wasserstein and Lazar, 2016). For BOU, CAR and PAY, IMF6 accounts for the yearly variability of the time-series, hence the correlation matrices are $6 \times 6$ in size. For TAT, the
yearly mode is IMF7, thus in this case the correlation matrix has a size of $7 \times 7$. Two conclusions can be drawn from figure 10.

    Values of $\tau$ significantly different from zero, shown in red, are recorded in the last column, for all stations. These demonstrate a modulation of the amplitude of the components having sub-year time-scales, i.e. AM1...AM5, respectively AM6 for TAT, by the yearly IMF, at a statistically significant level ($p \sim 0$). The effect is most pronounced for PAY, as inferred from the darker red shades (larger rank correlation coefficients).

For the BOU and CAR datasets the first row (AM1) exhibits blue and dark blue cells for IMF3...IMF5 at the statistically significant level. This indicates a negative rank correlation. Similar, but lighter, amplitude modulation is observed on the second row (AM2), but only by IMF4 and IMF5. For the PAY series, this negative rank correlation is greatly reduced for the first row (light blue tones) and is absent in the second row. For TAT no such correlation can be observed. At this point it is interesting to note, that in a similar way to the discussion from section 5.2, the different features of the datasets from figure 10 also enable a
classification of the local climate experienced by the measuring stations.

    It should be mentioned that the amplitude modulation of high-frequency "noise-like" components by lower frequency ones is also found in sunspots number time series (Chen et al., 2013) and in multiple solar proxies (Kolotkov et al., 2015). The short term intrinsic periodicities in the solar proxies appear to be indicative of "randomly distributed dynamical processes in the solar atmosphere" that are closely related to the 11 year solar activity and therefore, unsurprisingly, the high-frequency modes are
found to be modulated by this latter cycle (Kolotkov et al., 2016). But this phenomenon is not limited to solar activity signals, and has also been identified in surface air temperature records (Paluš, 2014), and time-series of the sea level (Liu et al., 2007), and may indicate cross-scale non-linear couplings (Paluš, 2014; Huang et al., 2016). Following this train of thought, the term "weather noise" (see Chekroun et al., 2011) is adopted for the high-frequency band defined by the first five IMFs in the Hilbert spectra of the SSI data.





## 6 Conclusion and outlook

To sum up, the HHT analysis of decennial time-series of daily means of measurements of the SSI, from distinct BSRN stations has revealed the following: the presence of high-frequency (2-100 days) "weather noise" consisting of quasi-stochastic IMFs that have been shown to be amplitude-modulated by the yearly cycle; a low power spectral spectral band in the 100 days to 300 days region; a well-defined spectral peak at the one year mark accounting for the yearly variability; and multiple "QBO" components whose character has been, inconclusively, attributed to quasi-stochastic random processes.

This separation of the (quasi-)periodic components of the signal from the apparently random realizations of a noisy background has been shown to significantly augment accuracy in time-series modelling (Rios and de Mello, 2013). Our findings can be thus directly used to improve models for estimating SSI from satellite images or forecasts of the SSI.

We have shown that the adaptive Hilbert-Huang Transform (HHT) is a versatile tool in analysing SSI data-sets, exhibiting significant non-linearity and non-stationarity. First, we have employed it to extract the intrinsic modes of variability of the SSI at distinct time-scales. Second, the HHT has been used to discriminate between the deterministic yearly cycle and the quasi-stochastic "weather noise". The same methodology could also be employed on different geophysical signals, such as wind speed time-series, river discharge datasets, etc.

When modelling climate processes as dynamical systems with low-frequency oscillations and noise effects, Chekroun et al. (2011) have shown that "even the 'approximately right' noise can help, rather than hinder". Here, we have provided a recipe not only for extracting, but also for characterizing the "weather noise" constituents of long-term time-series of the SSI. Indications with respect to modelling these quasi-stochastic components have also been provided. With respect to SSI forecast models, it is exactly this "weather noise" component that is the focus of attention (Ehnberg and Bollen, 2005; Hoff and Perez, 2010; Marquez and Coimbra, 2013). Inman et al. (2013) venture as far as stating that "the accuracy of the solar irradiance forecasting models depends almost exclusively on the ability to forecast the stochastic component". In this light, the recipe for noise that we have put forth can be seen as one of the more significant contributions of our paper.

We have also proposed that a classification of the measuring stations according to climate and/or solar insolation conditions may be possible, based on the Hilbert spectral features of the data. Thus, one future research pathway could consist in creating a catalogue of the variability of the solar resource, at different time-scales, on a global scale via satellite estimates of the SSI. Current meteorological re-analyses are too noisy in their estimates of the SSI to form the basis for such a catalogue (Boilley and Wald, 2015). In terms of solar power production, the low-frequency variability data would aid with policy and investment decisions, while short-term variability would be of interest from a monitoring, operations and engineering perspective.

## 7 Code availability

The software used for this study, comprising general EMD and HSA routines is publicly available online, as follows:

- The fast EMD routine used in this study is provided by Wang et al. (2014) and can be downloaded at:
  http://rcada.ncu.edu.tw/FEEMD.rar



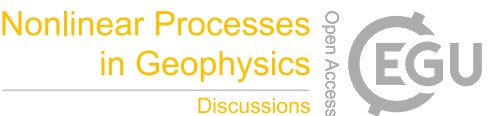

- Methods pertaining to Hilbert spectral analysis are part of a general HHT toolkit provided by Wu and Huang (2009) and can be downloaded at:

  http://rcada.ncu.edu.tw/Matlab%20runcode.zip

- The code for the ICEEMD(AN) algorithm (Colominas et al., 2014) is provided by María Eugenia Torres on her personal webpage, and can be downloaded at:

  http://bioingenieria.edu.ar/grupos/ldnlys/metorres/metorres_files/ceemdan_v2014.m

## 8   Data availability

The raw BSRN datasets employed in this study are made available by König-Langlo et al. (2015). Zip archives containing the data can be found at:

https://doi.pangaea.de/10.1594/PANGAEA.852720.

*Author contributions.*   All authors contributed equally to this work.

*Competing interests.*   The authors declare no competing interests.

*Disclaimer.*   N/A.

*Acknowledgements.*   The authors wish to thank Patrick Flandrin from Ecole Normale Supérieure de Lyon, France, and Gerard Thuiller from Laboratoire Atmosphères, Milieux, Observations Spatiales in Guyancourt, France, for the fruitful conversations and their insightful comments that have sparked the development of this study. Dmitrii Kolotkov from the University of Warwick, United Kingdom, is also acknowledged for the personal communications pertaining to the stochastic nature of the "weather noise" band."





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

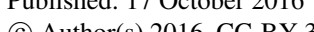
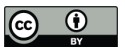
## List of Figures

## List of Tables





**Figure 1.** The four decennial SSI time-series investigated in this study, spanning 2001 through 2010. From top to bottom: BOU, CAR, PAY, and TAT. Each point corresponds to a daily mean of SSI. Time markers on the abscissa indicate the start of the corresponding year.





**Figure 2.** Histograms of the daily clearness index $K_T$ over the decennial time-span in frequency in percent. From top to bottom: BOU, CAR, PAY, and TAT.





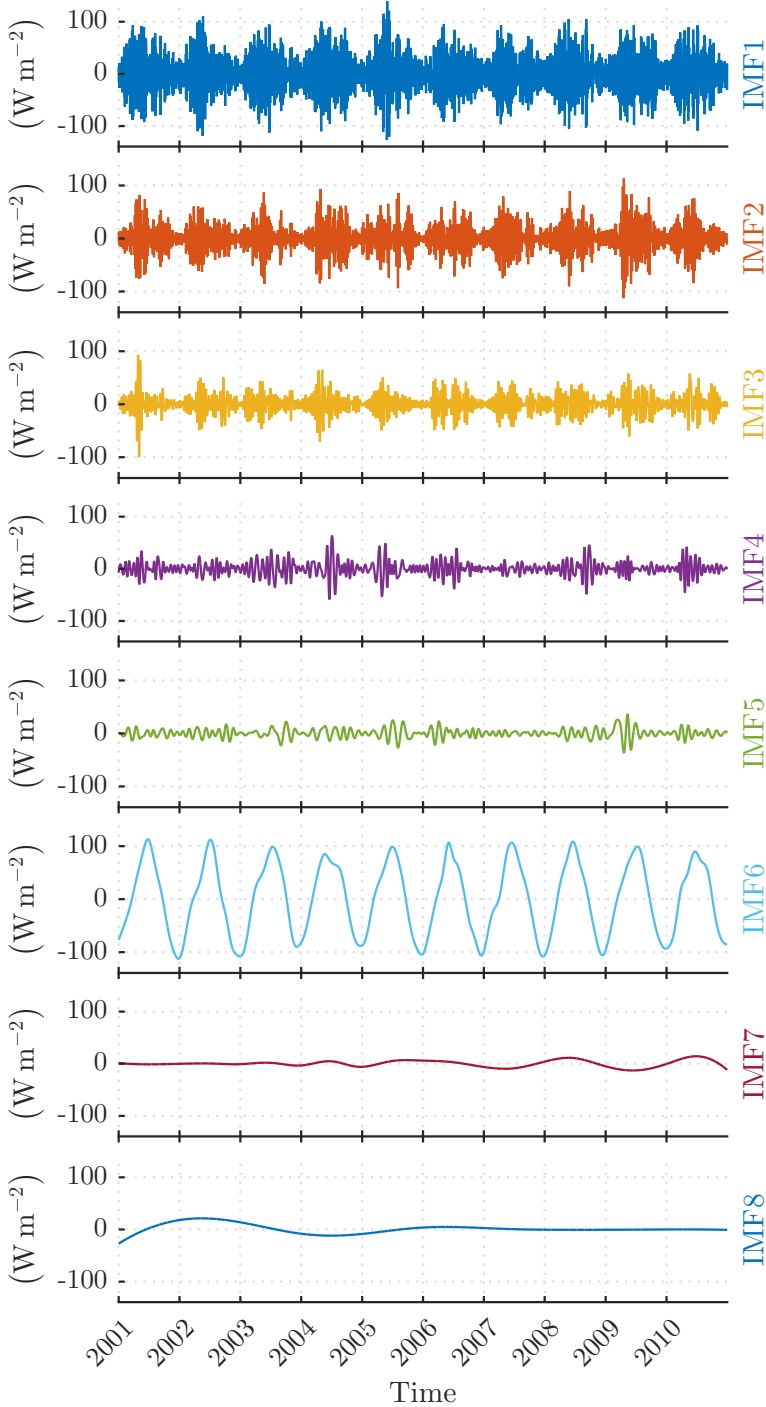

**Figure 3.** The eight IMFs obtained by decomposing the BOU time-series, from top to bottom IMF1…IMF8. The panels plot SSI (ordinate) versus time (abscissa). Time markers on the horizontal axes indicate January 1st of the corresponding year. The zero-centred oscillatory nature of the modes can be clearly seen. Also apparent is the local time-scale increase with mode number.





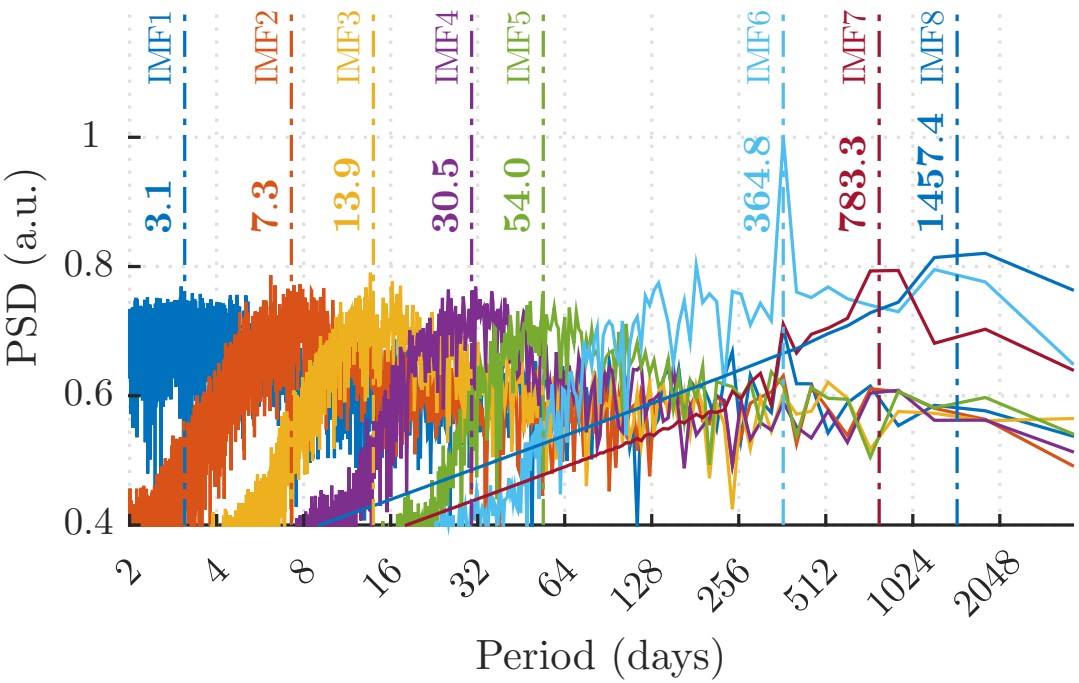

**Figure 4.** The power spectral density (PSD) of the eight IMFs for BOU (solid line) on a logarithmic scale normalized with respect to the power of the highest spectral peak. The period, or inverse frequency, runs on the abscissa in a base-2 logarithm. The individual spectra are shown in the same colours as the IMFs from figure 3, from left to right: IMF1...IMF8. The Fourier estimates of the median periods, marked along the dash-dotted lines, are seen to increase with mode number. Notable features are the prominent spectral peak of IMF6 at ∼ 365 days corresponding to the yearly cycle and the apparent dyadic repartition of the time-scales for IMF1...IMF5.



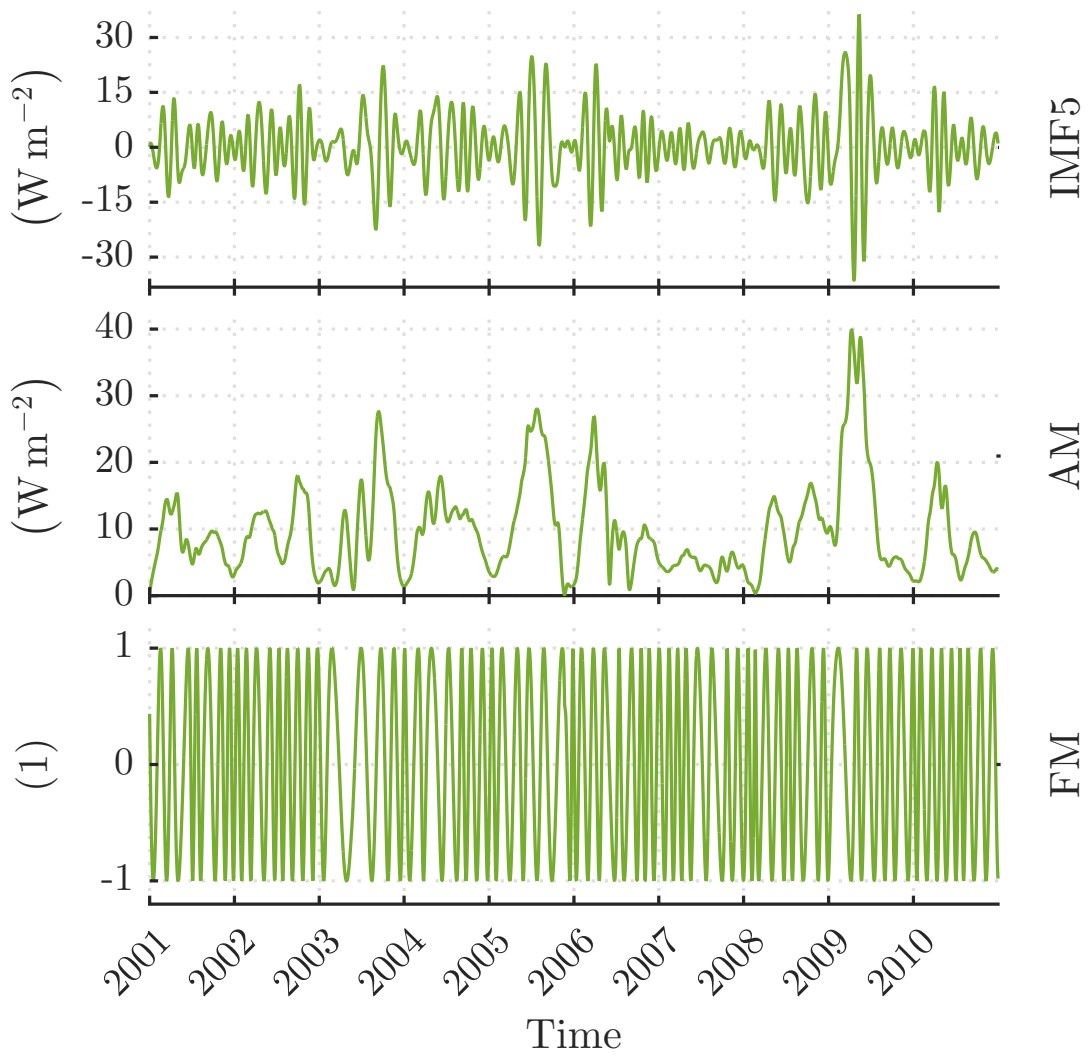

**Figure 5.** Hilbert spectral analysis of the fifth IMF of the BOU time-series. The intrinsic mode function (IMF5, top panel) is the product of its constituent slowly-varying amplitude modulation part (AM, middle panel) and of its rapidly-changing frequency-modulation component (FM, bottom panel). Time markers on the abscissa denote the beginning of the corresponding year.





**Figure 6.** The Hilbert spectrum (left panel) of the ten year time-series of SSI for BOU, spanning 2001 through 2010. Pixel colour encodes power (logarithmic scale colour bar on top) at each instant (abscissa) and each scale (ordinate). Time markers on the horizontal axis denote the start of the corresponding year. The white-out area indicates the regions where edge effects become significant. The Hilbert marginal spectrum in the right panel is the time-integrated version, i.e. line-by-line sum, of the Hilbert spectrum and indicates the amount of power at each scale.



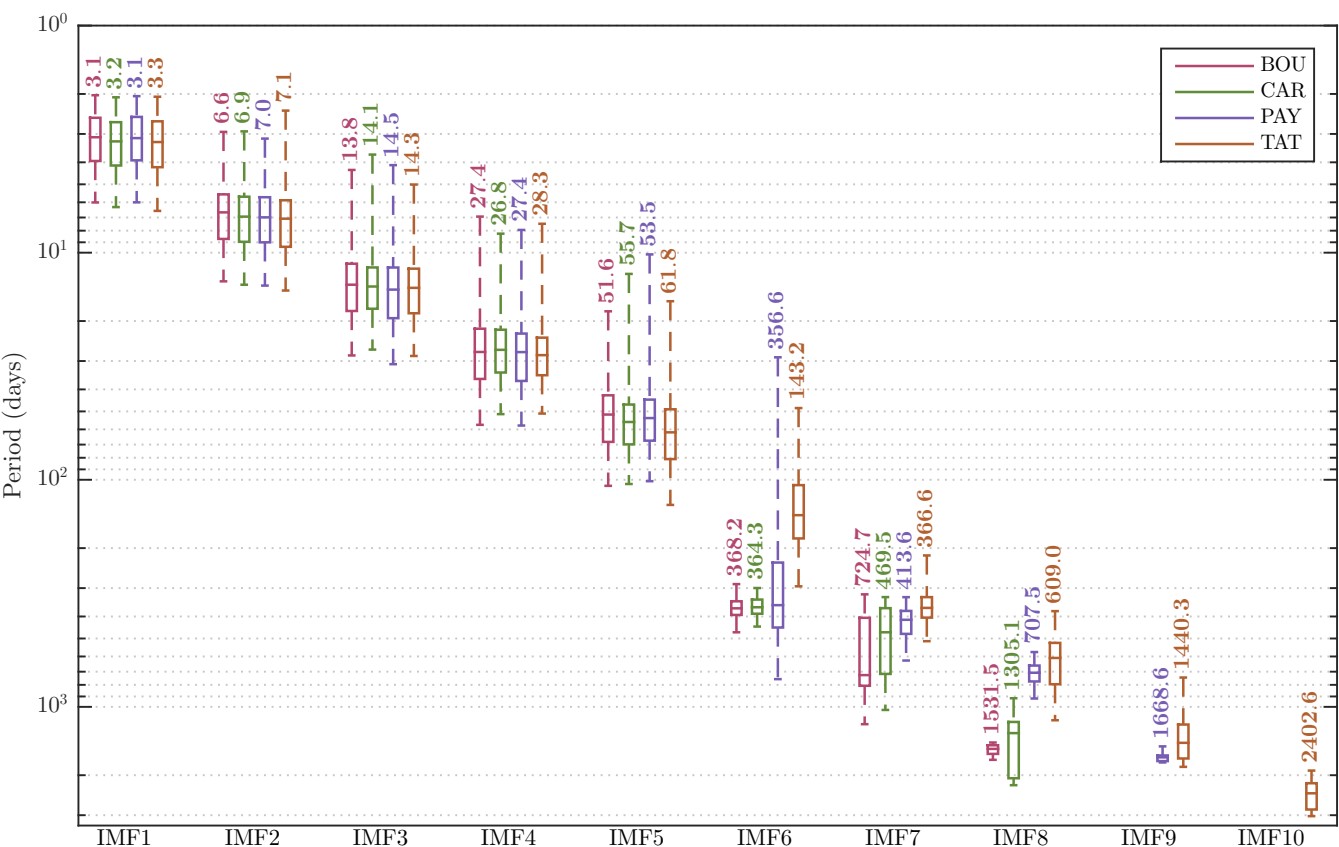

**Figure 7.** Box plot of the instantaneous time-scales of the IMFs for the four stations. The top and the bottom edges of the boxes represent the first ($Q1$) and, respectively, the third ($Q3$) quartiles. The bars inside boxes denote the second quartile ($Q2$), i.e. the median. The whisker length is set at at most 1.5 times the interquartile range, i.e. $1.5 \times (Q3 - Q1)$, hence the whiskers roughly correspond to $\pm 2.7$ standard deviations, or equivalently $\backsim 99\%$ of the data, assuming normal distribution. The median for each box is expressed numerically above the lower whiskers. Outliers are omitted.



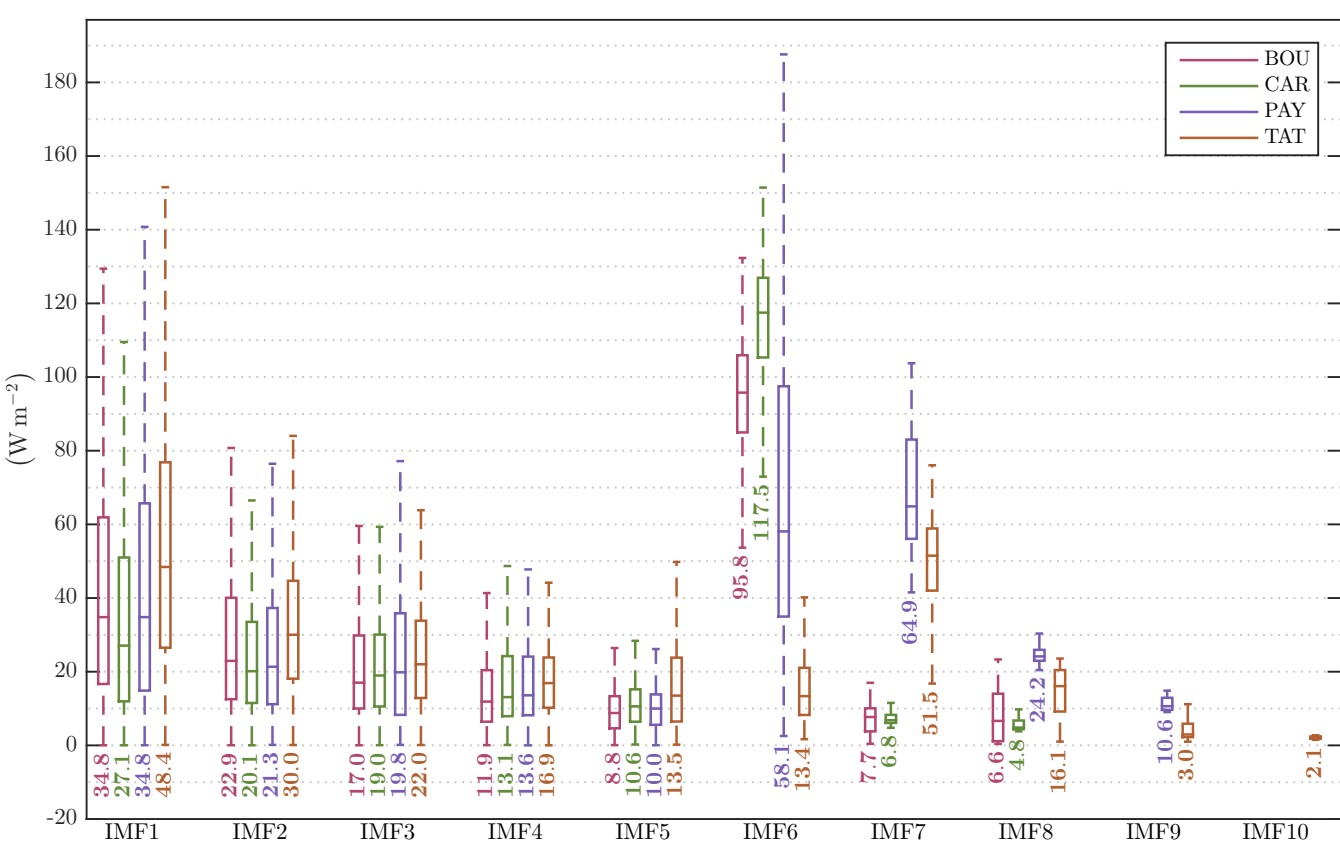

**Figure 8.** Box plot of the instantaneous amplitudes of the IMFs for the four stations. The bottom and the top edges of the boxes represent the first ($Q1$) and, respectively, the third ($Q3$) quartiles. The bars inside boxes denote the second quartile ($Q2$), i.e. the median. The whisker length is set at at most 1.5 times the interquartile range, i.e. $1.5 \times (Q3 - Q1)$, hence the whiskers roughly correspond to $\pm 2.7$ standard deviations, or equivalently $\backsim 99\%$ of the data, assuming normal distribution. The median for each box is expressed numerically below the lower whiskers. Outliers are omitted.





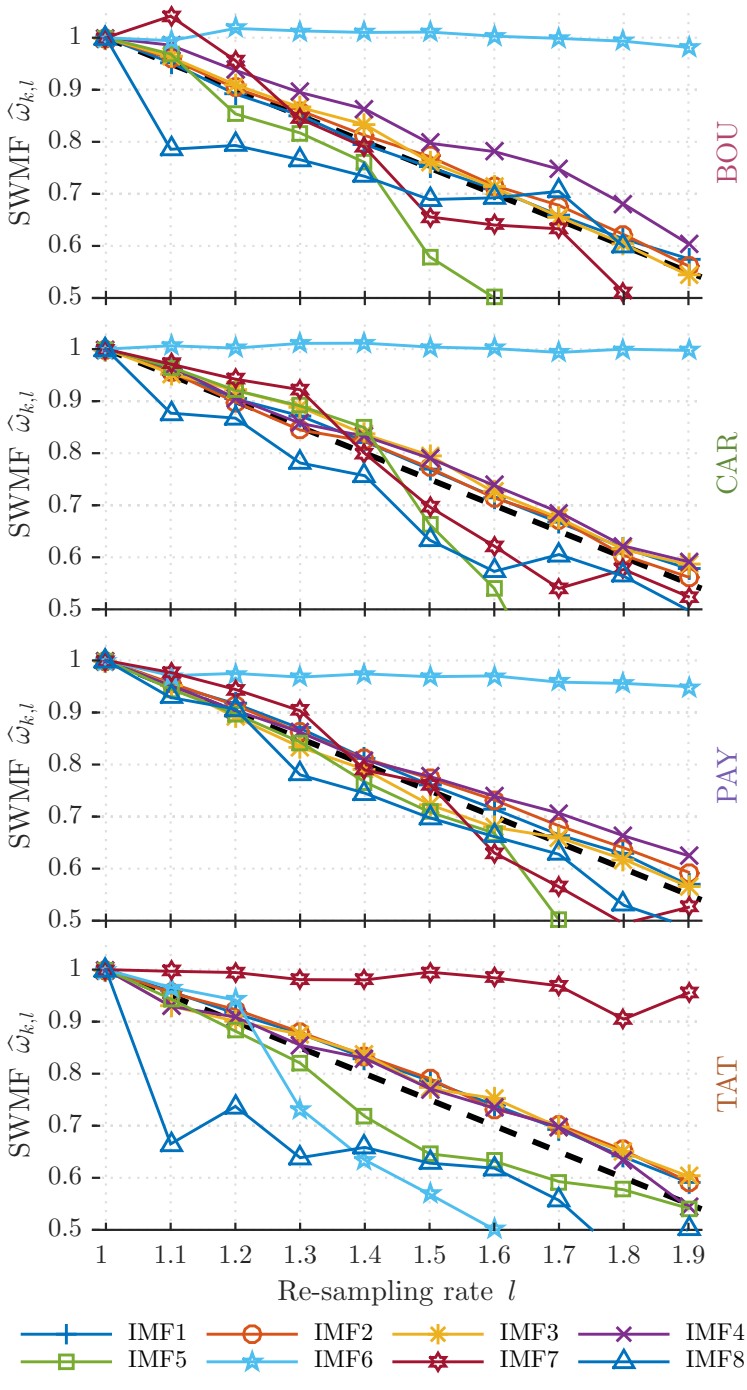

**Figure 9.** The drift of normalized SWMF $\widehat{\omega}_{k,l}$ (ordinate) of IMF $k$, with $k \in \{1 \ldots 8\}$, as a function of the re-sampling rate $l$ (abscissa) for the four time-series. From top to bottom: BOU, CAR, PAY, and TAT. The black dashed diagonal depicts the behaviour of a pure noise time-series under an ideal dyadic filter. For all datasets, the only mode that maintains a quasi-constant frequency under fractional re-sampling is the IMF associated with the yearly cycle, i.e. IMF6 for BOU,CAR, and PAY and IMF7 for TAT. In all the other IMFs quasi-stochastic behaviour is apparent, through frequency down-shifting towards the next lower octave, approximately following the dashed line.

**Figure 10.** Rank correlations between IMFs and their AM components for BOU (top left), CAR (top right), PAY (bottom left) and TAT (bottom right). Kendall's rank correlation coefficient $\tau$ is colour-coded according to the colour bar on the bottom. IMFs run vertically, along the columns, and their AM components run horizontally, along the rows. The numeric values within the cells are the associated $p$-values.



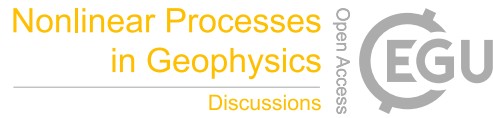

**Table 1.** Ground measurement stations listing.

| Code | Location* | | Latitude† | Longitude† | Climate‡ |
|------|-----------|------|-----------|------------|----------|
| BOU | Boulder | (US) | 40.0500 | -105.0070 | BSk |
| CAR | Carpentras | (FR) | 44.0830 | 5.0590 | Csa |
| PAY | Payerne | (CH) | 46.8150 | 6.9440 | Cfb |
| TAT | Tateno | (JP) | 36.0581 | 140.1258 | Cfa |

* Country codes according to ISO 3166-1 alpha-2
† Positive north for latitude and positive east for longitude, following ISO 19115
‡ Köppen-Geiger climate classification according to Kottek et al. (2006)