# Peer review of "On the intrinsic time-scales of temporal variability in measurements of the surface solar radiation"

_Nonlinear Processes in Geophysics, 2016_

## Referee Comment (RC1) · Anonymous Referee #1 · 28 Nov 2016

In this paper the authors consider the variability in the surface solar irradiance, using the empirical mode decomposition associated with the Hilbert-Huang transform. Globally I have no problem with this analysis, which is relevant and well described.

Only a major point needs to be explained and justified more clearly, or modified. This concerns a discrimination method between "signal" and "noise", first expressed lines 31-33 of page 2. The authors oppose deterministic information and stochastic variability, which is here considered as noisy background. This needs to receive a much more precise definition of terms, because it seems that, for the authors, something stochastic is purely noisy and not relevant for the physics of the problem studied. If this is correctly understood by the reader, it is not correct, since stochastic variability

possesses of course in general much more rich information than a pure noise. And second, because physical processes can generate stochasticity, such as e.g. turbulence. To oppose information and stochastic variability is not possible. The terms "weather noise" found in line 10, page 2, need to be renamed, since a noise has not correlation and weather have some. The procedure which is applied here to separate what is assumed to be "noisy" and deterministic information, is explained in section 3.2. The main idea is to state that "noisy" parts of the signal generate dyadic filtering in the EMD method, and a detection method based on this property is applied here. This is problematic because if white noise or fractional Brownian motion have been shown to generate EMD modes which are dyadic, the reciprocal is wrong, many studies have found the dyadic property for stochastic processes and also for observed data, that are not noises. The problem seems here the confusion by the authors between noise and random processes. The same confusion is visible in section 5.3, lines 17-19 and line 25. All this methodology and the discussion in section 5.3 must be changed or suppressed.

Another comment: the Hilbert-Huang marginal power spectrum of the data given by equation (8) should be displayed, for some locations and also globally.
* * *

---

## Short Comment (SC1) · 29 Nov 2016

**Review of the manuscript: "On the intrinsic time-scales of temporal variability in measurements of the surface solar radiation"**
**by M. Bengulescu, P. Blanc and L. Wald, NPGD, doi:10.5194/npg-2016-38**

**General comment**

The paper presents a study of the intrinsic temporal scales of the variability of the surface solar irradiance (SSI) by using a nonlinear and nonstationary method such as the empirical mode decomposition (EMD). The significance of the EMD results is also tested by using a novel and adaptive null-hypothesis test. The main result is the existence of a dominant spectral peak corresponding to the yearly variability cycle, a high-frequency "weather noise" and a low-frequency signature.

The paper is clearly written, logically organized and the results are well-presented. I recommend it for publication in Nonlinear Processes in Geophysics, once some minor comments and general features have been addressed.

**Major remarks**

1. I suggest to insert some explanations and discussions about the boundary effect problem and on the stopping criteria for the sifting process (pag. 5, Section 3.1.1). I think the adopted ones should be indicated.

2. Some additional descriptions on time variations observed in Figure 5 (bottom panel) could be very useful or, alternatively, a plot of the instantaneous frequency could be added to better show its time variations, also to visually facilitate the reader (pag. 7, line 5).

3. I think that the null-hypothesis test proposed in Section 3.2 is a simple but powerful test to investigate the noise-like existence of IMFs. If I do not misunderstood, this is particularly suitable when the EMD really acts as a dyadic filter. I suggest to remark this also when you describe Figure 7 in which a "dyadic" behavior can be observed for the high-frequency modes.

4. The results discussed in Section 5.4 are really important in the framework of weather and climate systems study. Also a cross-phase analysis could be useful to support these findings (not only related to the AM but also with FM component).

**Minor remarks**

- Abstract, Line 2: remove "spanning ten years" since it is redundant

- Abstract, Line 4: what does it mean "roughly" here? The EMD effectively sorts the IMFs with their increasing timescales

- Pag. 2, Line 2: "measured in" should be "observed on"

- Pag. 2, line 22: please insert HHT abbreviation

- Pag. 2, line 26-27: some references regarding the application of the EMD to weather and climate systems, together with solar time series, could be useful to support your choice in these frameworks (see references below)

- Pag. 3, line 5: in the framework of denoising signals a method was proposed to discriminate high-frequency flucuations from large-timescale modulation (see Flandrin et al, 2004; Alberti et al, 2016)

- Pag. 5: really good explanation about the EMD sifting process, not the usual one

- Pag. 6, line 1: why did you not include the residue in figure 3?

- Pag. 6, line 3: this is not properly correct. As you explained before, an IMF is a function whose envelopes are symmetric with respect to zero and not of zero mean.

- pag. 7, Equation (7): for completeness, the integral should be a sum, since you have discrete time series

- Pag. 10, line 5: a visual inspection of figure 3 shows that only the first 3 IMFs seem to have a clear annual modulation

- Pag. 10, line 19: I suggest to insert a table with the characteristic periods of each IMF for the 4 data sets with the range of variability (this could be a benefit for the reader)

- Pag. 10, line 31: the transitional mode could be explained in terms of physical processes such as monsoon rainy seasonality?

- Pag. 13, line 6: I suggest to include some references about short-term solar rotiational periodicities and related terrestrial signatures (see Prabhakaran, 2006; Emery et al, 2011, Morner, 2013)

- Pag. 14, line 16: IMFs 6 should be IMF6?

- Pag. 16, line 19: I suggest to insert the background color below each matrix.

**Suggested references for the EMD**

Alberti, T., Lepreti, F., Vecchio, A., Bevacqua, E., Capparelli, V. and Carbone, V.: Natural periodicities and Northern Hemisphere-Souther Hemisphere connection of fast temperature changes during the last glacial period: EPICA and NGRIP revisited, Clim. Past, 10, 1751-1762, 2014

Chambers, D. P.: Evaluation of empirical mode decomposition for quantifying multi-decadal variations and acceleration in sea level records, Nonlin. Processes Geophys., 22, 157-166, 2015

Lee, T. and Ouarda, T.B.M.J.: Long-term prediction of precipitation and hydrologic extremes with nonstationary oscillation processes, J. Geophys. Res., 115, D13107, 2010

Lee, T. and Ouarda, T. B. M. J.: Prediction of climate nonstationary oscillation processes with empirical mode decomposition, J. Geophys. Res., 116, D06107, 2011

Lee, T.,and Ouarda, T. B. M. J.: Stochastic simulation of nonstationary oscillation hydroclimatic processes using empirical mode decomposition, Water Resour. Res., 48, W02514, 2012

Solé, J., Turiel, A. and Llebot, J.E.: Using empirical mode decomposition to correlate paleoclimatic time-series, Nat. Hazards Earth Syst. Sci., 7, 299-307, 2007

Terradas, J., Oliver, R. and Ballester, J.L.: Application of statistical techniques to the analysis of solar coronal oscillations, Astrophys. J., 614, 435-447, 2004

Vecchio, A., Capparelli, V. and Carbone, V.: The complex dynamics of the seasonal component of USA's surface temperature, Atmos. Chem. Phys., 10, 9657-9665, 2010

Wu, Z., Schneider, E.K., Kirtman, B.P., Sarachik, E.S., Huang, N.E. and Tucker, C.J.: The modulated annual cycle: an alternative reference frame for climate anomalies, Clim. Dynam., 31, 823, 2008

Zhen-Shan, L. and Xian, S.: Multi-scale analysis of global temperature changes and trend of a drop in temperature in the next 20 years, Meteorol. Atmos. Phys., 95, 115-121. 2007

**Other references**

Alberti, T., Piersanti, M., Vecchio, A., De Michelis, P., Lepreti, F., Carbone, V. and Primavera, L.: Identification of the different magnetic field contributions during a geomagnetic storm in magnetospheric and ground observations, Ann. Geophys., 34, 1069-1084, 2016

Emery, B.A., Richardson, I.G., Evans, D.S. et al.: Solar Rotational Periodicities and the Semiannual Variation in the Solar Wind, Radiation Belt, and Aurora, Sol. Phys., 274, 399-425, doi:10.1007/s11207-011-9758-x, 2011

Flandrin, P., Goncalves, P. and Rilling, G.: Detrending and Denoising with empirical mode decomposition, Proceedings of Eusipco, Wien, Austria, 2004

Morner, N.-A., Solar Wind, Earth's Rotation and Changes in Terrestrial Climate, Physical Review & Research International, 3(2):117-136, 2013

Prabhakaran Nayar, S.R.: Periodicities in solar activity and their signature in the terrestrial environment, ILWS WORKSHOP 2006, GOA, FEBRUARY 19-24, 2006

---

## Referee Comment (RC2) · Anonymous Referee #2 · 13 Jul 2017

The authors perform a careful EMD and Hilbert analysis of four sets of surface solar irradiance data. The analysis methods are clear, concise, and carefully executed. The authors find a statistically significant IMF with a period of ∼1yr and higher-frequency components that they term 'weather noise'. These results are described clearly in the abstract and conclusions sections, however, I believe sections 5.1 and 5.2 of the discussion sections may, in some respects, be misleading. I believe the paper is of sufficient quality to warrant publication. The figures are informative and well described in the text. However, I have a few suggested modifications that I believe would improve the paper prior to publication.

Primarily I believe it is important to establish the signal/noise status of the components before discussing their physical origin i.e. sections 5.3 and 5.4 should be placed before sections 5.1 and 5.2. These sections then question the validity of linking the various components to features observed in solar data e.g. the discussion of the high-frequency components with solar rotation, which appear to be due to noise and the dyadic properties of EMD.

Along the same lines in Section 5.3 it is stated that 'unambiguous interpretations of QBO-like components seems to be out of reach' and yet the authors still discuss the possibility that it could be related to the solar QBO.

If the authors insist on including this discussion I believe the terrestrial QBO should also be mentioned as this also has a well know impact on weather on Earth, such as the severity of winters, which would also affect cloud cover. However, it is my opinion that the authors should either not try and make any conclusions concerning the QBO or at least stress that with the current analysis they cannot be sure that this is a real signal.

Finally with regards to the QBO I believe that the link between galactic cosmic rays and cloud coverage is still highly debated and so I would either remove the comment concerning this or refer to papers concerning the debate.

Minor comments: P7, line 7 'Unlike in the Fourier decomposition, the amplitude is not a constant, but rather a time-dependent function.' Fourier decomposition doesn't imply the amplitude is constant it just doesn't provide any information on the time variation of the amplitude.

P10, l24: 'the only minor difference being the slightly extended range for TAT of 200 days as opposed to 300 days for the other two': 300 for TAT and 200 for the other two?

P13, l27: containt -> contain

P15, l18 – Is there a way of quantifying how far from unity \omega must go before the

null hypothesis can no longer be rejected?

Interactive
comment

---

## Author Comment (AC1) · 28 Sep 2017

On the intrinsic time-scales of temporal variability in measurements of the surface solar radiation"
by M. Bengulescu et al.

ANSWER to Tommaso Alberti

We thank Mr Tommaso Alberti for the review and encouraging comments.

COMMENT. *"I suggest to insert some explanations and discussions about the boundary effect problem and on the stopping criteria for the sifting process (pag. 5, Section 3.1.1). I think the adopted ones should be indicated."*

ANSWER. Section 3.1.1 has been amended to take into account this comment by discussing both the boundary effect problem and the stopping criteria.

COMMENT. *"Some additional descriptions on time variations observed in Figure 5 (bottom panel) could be very useful or, alternatively, a plot of the instantaneous frequency could be added to better show its time variations, also to visually facilitate the reader (pag. 7, line 5)."*

ANSWER. A plot of the instantaneous frequency was added in fig. 5.

COMMENT. *"I think that the null-hypothesis test proposed in Section 3.2 is a simple but powerful test to investigate the noise-like existence of IMFs. If I do not misunderstood, this is particularly suitable when the EMD really acts as a dyadic filter. I suggest to remark this also when you describe Figure 7 in which a "dyadic" behavior can be observed for the high-frequency modes."*

ANSWER. This was already mentioned w.r.t. the shapes of the Fourier spectra in figure 4. It is now done also w.r.t. figure 7.

COMMENT. *"The results discussed in Section 5.4 are really important in the framework of weather and climate systems study. Also a cross-phase analysis could be useful to support these findings (not only related to the AM but also with FM component)."*

ANSWER. Thank you for this comment. We agree that such an analysis would be of valuable support. We have had the chance to have a seminar from Dr Milan Palus from the Czech Academy of Sciences the day before the PhD defense by Marc Benguslescu in July 2017. Milan Palus is studying this amplitude modulation of high-frequency "noise-like" components by lower frequency ones using different methods than ours. We have evoked the possibility of joining forces. Unfortunately we could not afford such a study for the time being as this would have requested additional work which was fully out of reach in the present conditions, especially as the most skilled author (Marc Bengulescu) has left MINES ParisTech for a very demanding job in another domain. Anyway, this is still an open question at MINES ParisTech.

Paluš, M.: Multiscale atmospheric dynamics: cross-frequency phase-amplitude coupling in the air temperature, Physical review letters, 112,
078 702, doi:10/bhpj, 2014.

MINOR REMARKS.
Thank you for spotting these points. All of them have been taken into account – see text.

Answers to particular remarks.

*Page 6, line 1: why did you not include the residue in figure 3?*
This has been already explained in the first paragraph of the Result section. we have rephrased to make it clearer.

*Page 6, line 3: this is not properly correct. As you explained before, an IMF is a function whose envelopes are symmetric with respect to zero and not of zero mean.*
Yes, you are right. Corrected

*Page 7, Equation (7): for completeness, the integral should be a sum, since you have discrete time series*
Yes, you are right. Corrected

*Page 10, line 5: a visual inspection of figure 3 shows that only the first 3 IMFs seem to have a clear annual modulation*
Yes, we agree that the first 3 IMFs always exhibit a clear annual modulation. The text refers to a discussion of Fig. 4 and mentions in passing that in 2005 this modulation affects also the IMFs 1 to 5 in Fig. 3. We have slightly rephrased this part.

*Page 10, line 19: I suggest to insert a table with the characteristic periods of each IMF for the 4 data sets with the range of variability (this could be a benefit for the reader)*
Done

*Page 10, line 31: the transitional mode could be explained in terms of physical processes such as monsoon rainy seasonality?*
This could be tempting. We have not investigated this in depth. This could be contradicted by one result in the paper Bengulescu et al. (2017), where a similar transitional mode was found at Vienna (Austria, 159 days) and Kishinev (Moldova, 199 days), two locations which experience very different climates compared to Tateno.
Bengulescu, M., Blanc, P., Boilley, A., and Wald, L.: Do modelled or satellite-based estimates of surface solar irradiance accurately describe its temporal variability?, Advances in Science and Research, 14, 35-48, doi:10.5194/asr-14-35-2017, 2017.

*Page 13, line 6: I suggest to include some references about short-term solar rotational periodicities and related terrestrial signatures (see Prabhakaran, 2006; Emery et al, 2011, Morner, 2013)*
Done

*Page 16, line 19: I suggest to insert the background color below each matrix.*
Done

---

## Author Comment (AC2) · 28 Sep 2017

NPG-2016-38. On the intrinsic time-scales of temporal variability in measurements of the surface solar radiation" by M. Bengulescu et al.

ANSWER to Anonymous Referee #1

We thank the Anonymous Referee #1 for the review and encouraging comments.

MAJOR COMMENT by Referee #1: *"Only a major point needs to be explained and justified more clearly, or modified. This concerns a discrimination method between "signal" and "noise""*

ANSWER. We understand that we have created a confusion between "noise" and "stochastic component". This is spread throughout the text, including the term "weather noise" that we have borrowed from other authors (Chekroun, M. D., Kondrashov, D., and Ghil, M.:, Predicting stochastic systems by noise sampling, and application to the El Niño-Southern Oscillation, Proceedings of the National Academy of Sciences, 108, 11 766–11 771, doi:10/bpt5kk, 2011). In their work on ENSO forecasting, these authors model the ENSO as a climate signal (slowly varying signal) influenced by fast (i.e. rapidly varying) processes that they called weather noise. This terming was certainly appropriate in their case but not in ours. Adopting this terming has created confusion somehow which is the basis of most of the following comments made by the Reviewer.

To remove this confusion, we have rewritten the text in some parts to avoid the use of "weather noise" and to make a clear distinction between the stochastic component created by physical processes and noise.

COMMENT. *"This needs to receive a much more precise definition of terms, because it seems that, for the authors, something stochastic is purely noisy and not relevant for the physics of the problem studied. If this is correctly understood by the reader, it is not correct, since stochastic variability possesses of course in general much more rich information than a pure noise."*

ANSWER: We are fully aware of this and we have clearly stated in Section 5.3: "It will be subsequently shown that, for the first five IMFs at least, this is indeed the case; although (quasi-)stochastic in nature, they are not completely devoid of information."

COMMENT on section 3.2. *"The procedure which is applied here to separate what is assumed to be "noisy" and deterministic information, is explained in section 3.2. The main idea is to state that "noisy" parts of the signal generate dyadic filtering in the EMD method, and a detection method based on this property is applied here. This is problematic because if white noise or fractional Brownian motion have been shown to generate EMD modes which are dyadic, the reciprocal is wrong, many studies have found the dyadic property for stochastic processes and also for observed data, that are not noises. The problem seems here the confusion by the authors between noise and random processes."*

ANSWER: We are well aware of this. This is why, in the original section 5.3, we have stated: "At this point, several precautionary notes are compulsory. First, the rule of inference used here is *modus tollens*, i.e. the results from figure 9 do not imply that the modes who experience down-shift in their SWMFs are made up of pure noise."

In addition to our answer, we may mention a comment made in the interactive discussion by Dr. Tomasso Alberti, who fully agrees and even appreciates our approach. He wrote *"I think that the null-hypothesis test proposed in Section 3.2 is a simple but powerful test to investigate the noise-like existence of IMFs. If I do not misunderstood, this is particularly suitable when the EMD really acts as a dyadic filter."*

COMMENT on section 5.3: *"The same confusion is visible in section 5.3, lines 17-19 and line 25. All this methodology and the discussion in section 5.3 must be changed or suppressed"*

ANSWER. The Section 5.3 "Discriminating signals from noise in the IMFs" has an interest because of the underlying question of the significance of each IMF. It is important to ascertain whether an IMF results from a physical phenomenon, possibly of stochastic nature, or from noise. This section has been reformulated to avoid the confusion mentioned by the Reviewer. We have also changed the title accordingly; it is now "Discriminating deterministic signals from stochastic components in the IMFs".
Please, take also note that following suggestions by Reviewer #2, the Section 5.3 is now Section 5.1.

COMMENT. *"the Hilbert-Huang marginal power spectrum of the data given by equation (8) should be displayed, for some locations and also globally"*

ANSWER. This comment is partly unclear to us as we have not used the term "Hilbert-Huang marginal power spectrum". Eq. 8 defines the Hilbert marginal spectrum which is displayed for one (BOU) of the four studied locations.
We read this comment as a suggestion to add three Figures similar to Figure 6. If we do this, we are facing a major increase in the length of the paper. Displaying the four Figures needs room and they should be accompanied by comments. To illustrate this, we have excerpted 4 pages (pages 57-60) from the PhD thesis of Marc Bengulescu (defended in July 2017) that comprise the four suggested graphs (Fig. 4.3) and the associated comments. These pages are reproduced here. Please, skip the first paragraph of the first page.

With the scrutiny of the these low frequency components, the discussion of the time-scale distribution of the IMFs from figure 4.1 can now be concluded. However, as previously mentioned, this particular illustration, although instructive, is incomplete. First, the box plot representation does not take the instantaneous variations of frequency into account, but renders global aggregates instead – much like the traditional Fourier methods, with the interquartile range spread in addition. This is done on purpose, with the intent of making it easier for the readership not accustomed to the HHT to create analogies with the more familiar methods (e.g. Fourier analysis, wavelets, etc.). Second and last, this particular representation is totally devoid of any information pertaining to the local amplitude, or power, or variance, of the data. With these consideration in mind, the Hilbert spectra of the data, a true time-frequency representation for non-linear and non-stationary data, will be discussed next.

Figure 4.3 depicts the Hilbert spectra of the four BSRN time-series: BOU, CAR, PAY and TAT. These spectra are similar to the one already introduced in figure 3.11, with the exception that the Hilbert marginal spectrum is no longer expressed in decibels.

The BOU Hilbert spectrum from the top left panel of figure 4.3 exhibits a high-frequency feature between 2 days and ⌣ 100 days, which corresponds to the first five IMFs of the time-series. The instantaneous time-scales of these modes overlap (figure 4.1), hence the appearance on the Hilbert spectrum of a continuum instead of distinct bands. This spectral feature has relatively low power, that decreases with increasing period, as can be inferred from the sloped dent in the marginal Hilbert spectrum corresponding to this region. In the 2 days to 32 days band, amplitude modulation by the yearly cycle can be inferred from the periodic change in color, with yellow-green tones, occurring mostly during the high irradiance regime of summer, that turn blue during the winterly minima. Next, in the band between 100 and 300 days, a gap in the spectrum is apparent, as can also be inferred from the lack of support in this region for any of the BOU IMFs in figure 4.1. The yellow trace, corresponding to IMF6, exhibits frequency modulation around the one year period, seen as oscillations in the range of 300 to 450 days, which is also the support of this mode in the box plot of IMF time-scales. The colour of this IMF indicates that it has the highest power of all the components, as can also be inferred from the large peak on the marginal spectrum. The corresponding time-scale fluctuations are centred in 365 days, and frequency modulation is greatest during 2003 through 2005. From 2006 onwards, however, frequency modulation is less pronounced – perhaps capturing the low solar activity around the 2008 minimum in the eleven year cycle solar cycle [Hathaway, 2015]. The final two low-frequency, blue-green traces on the spectrum correspond to IMF7 and IMF8. For IMF7, mode mixing is apparent through the occasional sharing of the yearly time-scale band with IMF6, between mid-2003 and 2005. IMF7 has such low power that it fails to leave an imprint on the marginal spectrum and it seems to suddenly spring

[Figure]

**Figure 4.3** The Hilbert energy spectra of the 10-year BSRN time-series, spanning 2001 through 2010; clockwise from top left: BOU, CAR, TAT, PAY. Pixel colour encodes power (logarithmic scale colour bar at the bottom) at each time (abscissa) and each scale (ordinate). Time markers on the abscissa denote the start of the corresponding year. The white-out area indicates the regions where edge effects become significant. The Hilbert marginal spectra in the panels on the right indicate the amount of power at each scale.

into existence during summer 2003, i.e. it has negligible amplitude during the first two and a half years. IMF8 starts out in light-green hues and slowly vanishes around 2007. This last mode for BOU has a median period of 1500 days (see figure 4.1), with most of its power lying within edge effect territory; interpretation of this feature is thus ambiguous at best.

Referring to the CAR Hilbert spectrum, depicted in the upper right panel of figure 4.3, some features can be identified. A high-frequency plateau between 2 days and $\backsim$ 100 days is notable, corresponding to first five IMFs of the time-series. As can be seen in figure 4.1, the supports of these modes overlap, thus the appearance on the Hilbert spectrum of a contiguous plateau, as is the case for BOU. Also similar to BOU, the power of this feature

is low, manifested by a slight indentation on the marginal Hilbert spectrum for this band. A cyclic shift in color can also be observed in this band, which points to an amplitude modulation mechanism linked to the seasonal cycle – darker blue tones during winter turn yellow-green during summer. Similar to BOU, between 100 and 300 days a gap in the spectrum is again manifest, as hinted at by the time-scales of the CAR IMFs in figure 4.1, which do not cover this band. IMF6, in the form of the yellow trace oscillating around the one year period, in the range of 300 to 450 days, resembles the same mode for BOU. This component has the greatest power, denoted by the large peak on the marginal spectrum. It exhibits frequency modulation around 365 days, with shifts towards greater frequencies taking place predominantly during 2002 through 2006. Between 2007 and 2009, however, the frequency modulations are less pronounced – corresponding to a period when solar activity is at a minimum. The last two large time-scale, blue-green traces denote IMF7 and IMF8. The seventh exhibits some mode mixing with IMF6, between the end of 2003 and the first half of 2006, and shortly again before 2009. These two last components have such low power that they fail to leave a mark on the marginal spectrum.

In the Hilbert spectrum for the PAY data, shown in the lower left panel of figure 4.3, the contiguous high-frequency plateau between 2 days and ⌣ 100 days is also present, unsurprisingly, since the first five IMFs of the time-series closely resemble those for the CAR data. The power in this band, however, is slightly greater than for BOU or CAR, especially considering the predominance of yellow hues in the $2-4$ days region that corresponds to the first and second IMFs. This is also apparent when looking at the marginal spectrum, where a distinct peak at this time-scale can be clearly made out. The amplitude modulation phenomenon in phase with the seasonal variations, previously identified in the BOU and CAR data, is even more pronounced in the PAY spectrum; once again dark blue tones that occur during low insolation in winter turn green and even yellow during the high irradiance regime of summer. Here too, between 100 and 300 days a gap in the spectrum is also apparent, with the notable exception of the mode mixing phenomena associated with IMF6 that occur during 2003, 2007 and 2009. As previously shown in figure 4.1, the lower range of the frequency distribution of the sixth mode overlaps the high frequency plateau, which is portrayed by the three jutting spikes from the yearly band into the sub-100 days plateau on the PAY spectrum. Since these protruding filaments have such low power that they leave no imprint on the marginal spectrum, their most probable origin can be attributed to some sort of numerical artefacts. The yearly variability of the data can be, unsurprisingly, identified with the sharp peak at roughly 365 days on the marginal spectrum. In terms of the Hilbert time-frequency-energy representation, however, the seasonal cycle cannot be attributed to one sole component. This is due to mode mixing as seen in figure 4.1 where the range of IMF7 completely overlaps that of IMF6. It could be argued that IMF6 should represent the "true" seasonal cycle, as indicated by the median

of its frequency distribution, however for 2003 and 2007 the Hilbert spectrum reveals that whenever IMF6 extends its tendrils into the high-frequency plateau and drastically reduces its power, IMF7 seems to "pick up the slack" by reaching into the vacated one year band. During 2008, and briefly during spring 2004, it is found that the two components trade places altogether, with IMF6 assuming lower values that IMF7 on the frequency scale. The last two low-frequency components, IMF8 and IMF9, can be seen here too to have relatively low power – both only just manage to make a very slight indentation on the marginal spectrum. For IMF9, the frequency spread in the Hilbert representation is in good agreement with its narrow support from figure 4.1.

The Hilbert spectrum for the TAT data, shown in the lower right panel of figure 4.3, also shows the contiguous high-frequency plateau between 2 days and $\smile$ 100 days, owing to the first five IMFs of the data closely resembling those for BOU, CAR and PAY data. However, the power found in this band is much greater than for either BOU, CAR, or PAY, especially in the 2 – 4 days region denoting the first two IMFs. For these time-scales of TAT, the marginal spectrum shows a peak that is even greater than the one associated with the yearly cycle. This feature is also evident in figure 4.2, where the upper whisker of the amplitude of IMF1 is seen to extend beyond 150 $\mathrm{W\,m^{-2}}$, whereas it is less than 80 $\mathrm{W\,m^{-2}}$ for annual cycle depicted by IMF7. As a result, the slanting of the high-frequency plateau is clearly visible on the marginal spectrum, with a somewhat lesser slope that for the other stations. The amplitude modulation phenomenon in phase with the seasonal variations, also identified for the other datasets, is present here too, although to a lesser extent than for PAY. Unlike the previous datasets, no gap can is evident in the spectrum, owing to a sixth IMF that covers the region between 50 and 300 days, as shown in figure 4.1. The yearly variability of the time-series is denoted by the spectral peak at roughly 365 days on the marginal spectrum. In the Hilbert time-frequency representation the seasonal cycle is attributed to the dark yellow trace of IMF7, which can be seen in figure 4.1 to have a median time-scale of 366.6 days. Mode mixing in the yearly band is nevertheless apparent between the second half of 2005 through 2007, when IMF8 approaches the 365 days mark while IMF7 protrudes below 256 days. The last two low-frequency components, IMF9 and IMF10, have reduced power and fail to leave a mark on the marginal spectrum.

So far, all time-series have been shown to share a high-frequency constituent between 2 days and 100 days composed of five IMFs with mean periods following a dyadic sequence, and an IMF around 365 days that captures the yearly variability. For BOU, CAR and PAY, a low power region can be found in the 100 days to 300 days band. Beyond the one year time-scale, the low-frequency variability in the 1.5 years to 6 years band is captured by another two (BOU and CAR) or three (PAY and TAT) components. The TAT data is the only time-series that has an IMF in the low power band between the high-frequency feature and the yearly cycle (median period 143.2 days).

---

## Author Comment (AC3) · 28 Sep 2017

We thank the Anonymous Referee #2 for the review and encouraging comments.

MAJOR COMMENT "Primarily I believe it is important to establish the signal/noise status of the components before discussing their physical origin i.e. sections 5.3 and 5.4 should be placed before sections 5.1 and 5.2. These sections then question the validity of linking the various components to features observed in solar data e.g. the discussion of the high frequency components with solar rotation, which appear to be due to noise and the dyadic properties of EMD."

[Figure]

ANSWER. We thank the Reviewer for this suggestion. We agree that it facilitates the reading of this Section. This was done. Note that we have renamed Section 5.1 in "Discriminating deterministic signals from stochastic components in the IMFs".

COMMENT. "Along the same lines in Section 5.3 it is stated that 'unambiguous interpretations of QBO-like components seems to be out of reach' and yet the authors still discuss the possibility that it could be related to the solar QBO. If the authors insist on including this discussion I believe the terrestrial QBO should also be mentioned as this also has a well know impact on weather on Earth, such as the severity of winters, which would also affect cloud cover. However, it is my opinion that the authors should either not try and make any conclusions concerning the QBO or at least stress that with the current analysis they cannot be sure that this is a real signal. Finally with regards to the QBO I believe that the link between galactic cosmic rays and cloud coverage is still highly debated and so I would either remove the comment concerning this or refer to papers concerning the debate."

ANSWER. This paragraph has been rewritten and is now: Lastly, the components indicative of low-frequency variability on time-scales greater than one year are discussed. The intrinsic time-scales found in these IMFs seem to match once more those pertaining to the so-called quasi-biennial oscillations that have been observed in solar activities and proxies with periodicities between 0.6 and 4 years (Bazilevskaya et al., 2015; Kolotkov et al., 2015; Vecchio et al., 2012), as well in meteorological data like Harrison (2008) who identifies a 1.68 year peak in cloud cover or high-latitude stratospheric temperatures and geopotential heights (Labitzke and Loon, 1988). Nevertheless, within the scope of the current analysis, the interpretation of these low frequency variability components as as a real, possibly QBO-like, signal is uncertain.

MINOR REMARKS. Thank you for spotting these points. All of them have been taken into account and the text was rewritten accordingly.

2016-38, 2016.